# Fourier Position Embedding:
# Enhancing Attention's Periodic Extension for Length Generalization

**Ermo Hua**[1]  **Che Jiang**[1]  **Xingtai Lv**[1]  **Kaiyan Zhang**[1]  **Youbang Sun**[1 2]
**Yuchen Fan**[1 3]  **Xuekai Zhu**[1 4]  **Biqing Qi**[† 3]  **Ning Ding**[† 1]  **Bowen Zhou**[† 1 3]
https://github.com/TsinghuaC3I/Fourier-Position-Embedding

## Abstract

Extending the context length of Language Models (LMs) by improving Rotary Position Embedding (RoPE) has become a trend. While prior works mainly address RoPE's limitations within attention, this paper uncovers the adverse effects on length generalization from nearly all parts of LMs. Using *Discrete Signal Processing* theory, we show that RoPE enables periodic attention by implicitly achieving *Non-Uniform Discrete Fourier Transform*. However, this periodicity is undermined by the spectrum damage caused by: 1) linear layers and activation functions; 2) insufficiently trained frequency components brought by time-domain truncation. Building on our observations, we propose ***Fourier Position Embedding (FoPE)***, which enhances attention's frequency-domain properties to improve both its periodic extension and length generalization. FoPE constructs *Fourier Series* and zero-outs the destructive frequency components, increasing model robustness against the spectrum damage. Experiments across various model scales and benchmarks show that, within varying context windows, FoPE maintains a more stable performance compared to other baselines. Several analyses and ablations bring further support to our method and theoretical modeling.

## 1. Introduction

Generation based on the information from long contexts is crucial for Language Models (LMs). However, LMs are

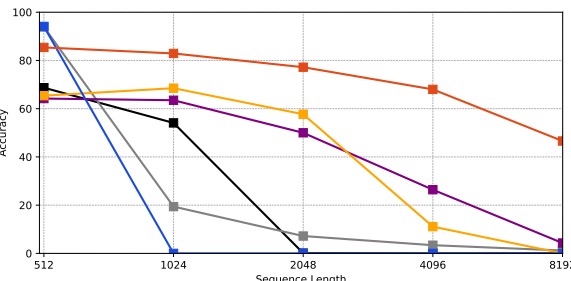

(a) Accuracy on Passkey Retrieval (higher is better)

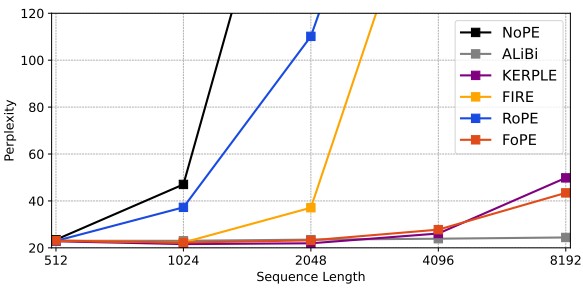

(b) Perplexity on C4 (lower is better)

*Figure 1.* Comparison of 1.2B models with different position embeddings (trained with a maximum sequence length of 512) on a needle-in-haystack task and pre-training perplexity. FoPE yields significantly more stable results than RoPE and other baselines.

typically trained on a fixed context window (Vaswani, 2017; Touvron et al., 2023; Groeneveld et al., 2024) and tends to overfit to the specific context length.

Many studies consider the absolute position embedding (Vaswani, 2017) as a key contributor to overfitting. As mitigation, several relative position embedding methods have been proposed (Press et al., 2021; Chi et al., 2022; Peng et al., 2023; Li et al., 2024; Jin et al., 2024; Su et al., 2024) to improve LMs' long-distance dependency. Among these, ALiBi (Press et al., 2021) introduced a position-biased attention mask, which linearly declines the attention weights based on distance. ALiBi achieves stable perplexity during pre-training, but fails to retain information from distant

---
[1]Tsinghua University [2]Northeastern University [3]Shanghai Artificial Intelligence Laboratory [4]Shanghai Jiaotong University. Correspondence to: Biqing Qi <qibiqing@pjlab.org.cn>, Ning Ding <dingning@tsinghua.edu.cn>, Bowen Zhou <zhoubowen@tsinghua.edu.cn>.

*Proceedings of the 42$^{nd}$ International Conference on Machine Learning*, Vancouver, Canada. PMLR 267, 2025. Copyright 2025 by the author(s).

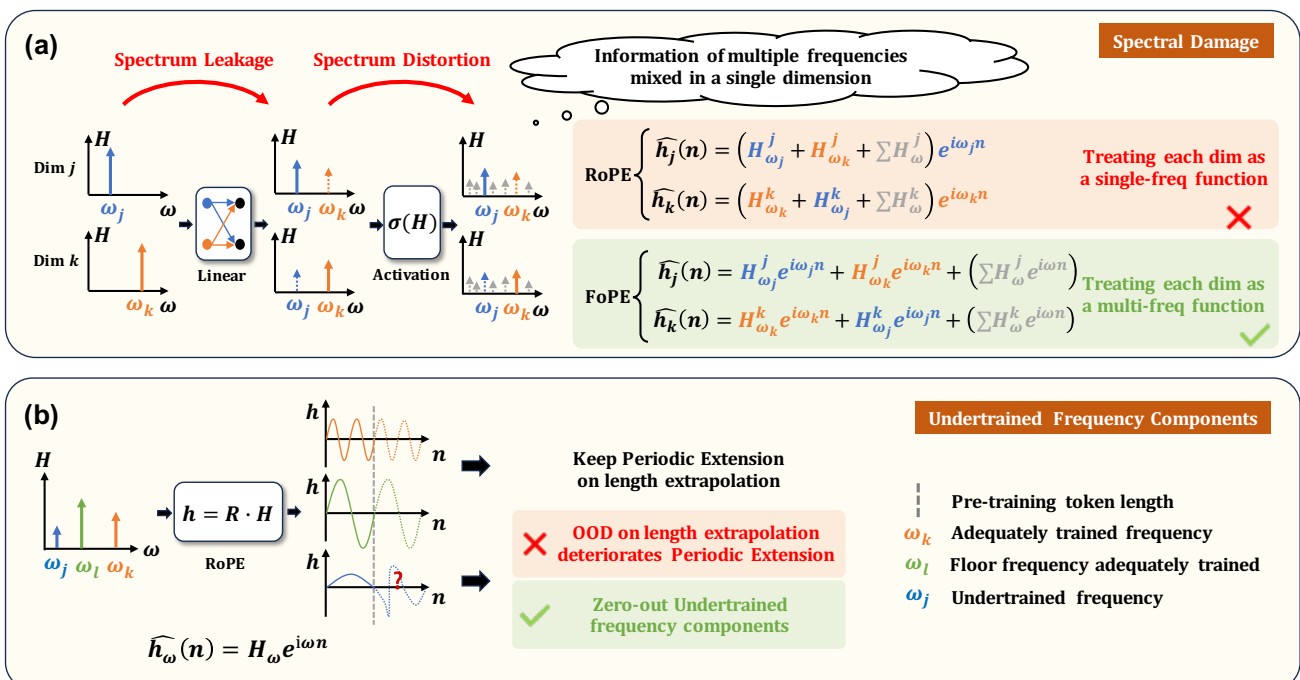

*Figure 2.* The reasons why RoPE's periodic extension deteriorates and how FoPE addresses these issues to improve length generalization. (a) As signals pass through linear and nonlinear transformations, this causes spectral leakage and distortion, mixing multiple frequencies into a single dimension. Under RoPE, each dimension is treated as a single-frequency component. By contrast, FoPE models each dimension as a Fourier series of different frequency components, thereby separating information more effectively and mitigating spectral damage. (b) FoPE eliminates inadequately trained frequency components, which are harmful for periodic extension. By preserving only the zero-frequency component, FoPE safeguards periodic extension and delivers more robust length generalization.

tokens, leading to poor performance on long-context downstream tasks. Another method, RoPE (Su et al., 2024), uses the phase of complex numbers to store the position information. Combined with continual pre-training and other interpolation-based methods (Peng et al., 2023; Xiong et al., 2024; Chen et al., 2024a; Jin et al., 2024), RoPE provides better access to long-distance information, making it one of the most widely used position embedding. However, RoPE-based LMs still struggle with length generalization without supplementary methods like YARN (Peng et al., 2023).

In this paper, we take a closer look at RoPE in the frequency-domain with tools from *Discrete Signal Processing (DSP)* theory. Our modeling reveals that RoPE implicitly performs *Non-Uniform Discrete Fourier Transform (NUDFT)* on the hidden states, enabling periodic attention based on the frequency-domain encoding. However, we find that the periodicity is hindered by the spectral damage caused by: 1) linear layers and activation functions outside attention; 2) inadequately-trained frequency components within attention (See Fig 2). This explains why RoPE fails to achieve length generalization without assistance from other methods.

Building on our observations above, we propose ***Fourier Position Embedding (FoPE)*** to further improve the attention's

periodic extension for better length generalization. Compared to RoPE, FoPE introduces two main improvements: 1) While RoPE treats each dimension as a single-frequency function, FoPE models each dimension as a *Fourier Series*, consisting of a dominate frequency component and several harmonic components. This approach better mirrors the actual spectrum in LMs and helps attention separate information across different wavelengths, mitigating the negative effects of Spectral Damage. 2) FoPE clips inadequately trained frequency components that is harmful to length generalization. To keep the passing of long wavelength information, we substitute these components with zero, as the zero-frequency component corresponds to the longest wavelength.

We summarize our contribution as follows:

1. Based on DSP, we provide frequency-domain analysis to reveal the negative influence from nearly all parts from LMs. We find that the length generalization is hindered by the Spectrum Damage arised from: 1) linear layers and activation functions; 2) undertrained frequency components.

2. We propose FoPE to improve attention's robustness on the Spectrum Damage. FoPE construct Fourier Series to extract multi-frequency information in each dimension, and

clip the frequency of destructive components to zero. Thus, FoPE delivers better periodic extension of attention, thus bringing better length generalizaion.

3. We conduct experiments across several model scales and datasets. The perplexity in pre-training and the accuracy in needle-in-haystack demonstrate FoPE's superiority over other baselines on length generalization. Evaluation on more complex tasks (i.e. summarization and few-shot question-answering) brings further support to our method and theoretical modeling.

## 2. Preliminaries

### 2.1. Non-Uniform Discrete Fourier Transform

Given a finite sequence of $\{x_n\} := x_0, x_1, ..., x_{N-1}$ equally-sampled from a continuous function $x$, **Discrete Fourier Transform (DFT)** converts them into equally-spaced components $\{X_m\} := X_0, X_1, ..., X_{M-1}$ in frequency-domain, the original samples can be recovered by Inverse DFT (IDFT):

$$X_m = \sum_{n=0}^{N-1} x_n e^{-i2\pi \frac{n}{N} m}, \quad x_n = \frac{1}{M} \sum_{m=0}^{M-1} X_m e^{i2\pi \frac{m}{M} n} \tag{1}$$

As $e^{i\omega n} = \cos \omega n + i \sin \omega n$ is periodic in the original domain, DFT implicitly transforms the original function into a linear combination of periodic waves with frequency $\omega_m = 2\pi \frac{m}{M}$. Thus, DFT is an estimation of the original function, which is lossless only if the original function exactly composes of these specific periodic components.

To achieve a more accurate approximation, the sampled frequencies $\omega_m := \omega_0, \omega_1, \ldots, \omega_{M-1}$ can follow any arbitrary distribution tailored to the data characteristics, subject only to the constraint $\omega_m \in [0, 2\pi)$. This generalized form of DFT is known as the **Non-Uniform Discrete Fourier Transform (NUDFT)**.

### 2.2. RoPE implicitly achieves Periodic Attention based on NUDFT

Given a $M$-dimension query $Q$ and key $K$ for tokens $a$ and $b$, RoPE rotates different dimensions $m$ to different phases:

$$\widetilde{q_m}(n_a) = Q_m e^{i\omega_m n_a}, \widetilde{k_m}(n_b) = K_m e^{i\omega_m n_b} \tag{2}$$

where $\omega_m = 1/\theta^{(2m/M)}$ and $\theta$ is the pre-defined parameters in RoPE. Then, the attention weight $h_m(n)$ in each dimension will be calculated as:

$$\widetilde{h_m}(n) = \widetilde{q_m}(n_a)\widetilde{k_m}^*(n_b) = H_m e^{i\omega_m n} \tag{3}$$

where $n = n_a - n_b$ and $H_m = Q_m K_m$. Finally, the overall attention weight between different tokens becomes:

$$h(n) = \sum_{m=0}^{M-1} \widetilde{h_m}(n) = \sum_{m=0}^{M-1} H_m e^{i\omega_m n} \tag{4}$$

Comparing it with Eq (1), it can be observed that RoPE implicitly achieves a token-level Inverse NUDFT with frequency components $\{\omega_m\}$.

Based on NUDFT, RoPE models the interactions between different tokens as functions compose of several periodic components, which brings **periodic extension** in each dimension $m$:

$$\widetilde{h_m}(n + N_{\omega_m}) = \widetilde{h_m}(n) \tag{5}$$

where $N_{\omega_m} = \frac{2\pi}{\omega_m}$ is this component's period. This property can generalize LMs to longer context.

## 3. Spectrum Damage Confine the Length Generalization

Ideally, RoPE-based Attention achieves periodic extension in any length scenario. However, this extension is confined as a key ideal property that is not guaranteed in LMs.

### 3.1. Negative Influence of Spectrum Damage

The ideal coefficients and frequencies of NUDFT have one-to-one correspondence. The coefficient of each frequency represents the influence of each token on others propagated at a specific wavelength.

However, the periodic extension is hindered, if the coefficient also contains the information from another frequency component $\omega_o$ with coefficient $H_{\omega_o} = \sigma H_\omega$, called the **Spectrum Damage**.

If we define the damaged function as $h'_m = H_{\omega_m}[(1 - \sigma)e^{i\omega_m n} + \sigma e^{i\omega_o n}]$, we find:

$$h'_m(n + N_{\omega_m}) \neq h'_m(n) \tag{6}$$

as $N_{\omega_m}$ is not the period of $h_{\omega_o}$. In other words, the information from each component is transmitted through waves with mismatched wavelengths, leading to inaccurate estimation of the influence propagated within each wavelength. (See Fig 3) When multiple frequency components are mixed within a dimension, the periodicity of information variation with token distance changes (e.g., Ground Truth in the top and middle sub-figure), no longer matching the periodicity expected by RoPE for that dimension. As a result, the periodic extension and length generalization of attention are adversely affected.

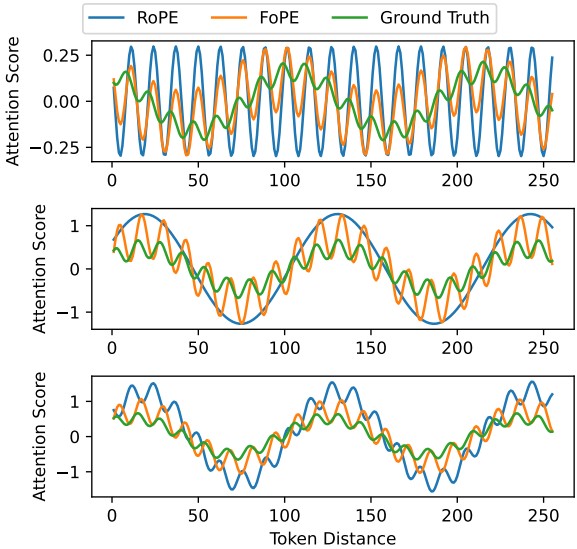

*Figure 3.* An illustration of the negative impact of Spectrum Damage on length generalization. The figure show the contributions of two frequency components to the final attention score (top & middle), followed by the sum of their contributions (bottom). We consider a two dimensions hidden states with two frequency components. Because of Spectrum Damage, each dimension actually contains multi-frequency information (referencing Ground Truth). The multi-frequency position embedding introduced by FoPE produces attention scores that are closer to the actual information compared to RoPE.

### 3.2. Spectrum Damage Outside Attention

The LMs' linear layers and activation functions outside attention bring two types of spectrum damage, destroying the one-to-one correspondence between coefficients and frequencies.

**Linear Layer** uses weights $W \in \mathbb{R}^{M \times M}$ to map a $M$ dimension hidden state $X \in \mathbb{R}^M$ to another hidden state $Y \in \mathbb{R}^M$. Thus, each dimension of $Y$ will be a linear combination of different components of $X$:

$$Y_m = \sum_{k=0}^{M-1} W_{km} X_k \qquad (7)$$

This results in ***Spectrum Leakage***, as different frequency components exhibit interplay.

**Activation Function** has non-linearity in the time domain, generating harmonic frequencies as described by the following Lemma:

**Lemma 3.1.** *Given a double-frequency sinusoid function $x(n) = \cos \omega_1 n + \cos \omega_2 n$ and any time-independent non-linear function $g$. The effect of $g$ on $x(n)$ yields functions*

*with frequencies as linear combinations of $\omega_1$ and $\omega_2$:*

$$g(x(n)) = \sum_{j \in N} \sum_{k \in N} a_{j,k} \cos(j\omega_1 + k\omega_2)n \qquad (8)$$

*which can be generalized to any multi-frequency function $x(n) = \sum(a_\omega \sin \omega n + b_\omega \cos \omega n)$[1].*

As the hidden states are transformed into multi-frequency functions by Linear Layer, passing them across Activation Functions introduces additional harmonic components, leading to serious ***Spectrum Distortion***.

These two types of Spectrum Damage undermine the periodic extension of attention (as shown in Eq.(5)(6)), hindering the model's length generalization property.

### 3.3. Spectrum Damage Inside Attention

Besides the spectrum damage outside attention, the undertrained components of attention within extremely low frequencies ($\omega_m < \frac{2\pi}{N}$) also bring spectrum damage.

Given a single-frequency function $x_m(n) = e^{i\omega_m n} \text{rect}(n)$ truncated by a square wave:

$$\text{rect}(n) = \begin{cases} 1 & , n \leq N \\ 0 & , n > N \end{cases} \qquad (9)$$

Based on the results of DFT, the spectrum of x(n) is[1]:

$$X(\omega) = \alpha \delta(\omega_m) + \frac{\sin[(N - \alpha N_m)(\omega - \omega_m)]}{\omega - \omega_m} \qquad (10)$$

where $\alpha = \lfloor \frac{N}{N_m} \rfloor$ and $N_m = \frac{2\pi}{\omega_m}$.

In the frequency domain, time-domain truncation introduces noisy components via the latter sub-function. When the period of the primary frequency component exceeds the truncation length, its amplitude is significantly weakened. Consequently, noisy components dominate these dimensions, impairing the periodic extension (as defined by Eq.(5)(6)). In contrast, high-frequency components are minimally affected because their coefficients $\alpha$ dominate over the noisy components.

Intuitively, when sampling sinusoidal functions based on token positions, these low-frequency components cannot cover a complete cycle. Therefore, for positions exceeding the pre-training sequence length, these dimensions may sample outside the training domain, leading to difficulties in generalization. Although previous works (Peng et al., 2023) have identified this issue, we are the first to model it from a Fourier perspective and provide a theoretical explanation.

---

[1]From (Oppenheim et al., 1982)

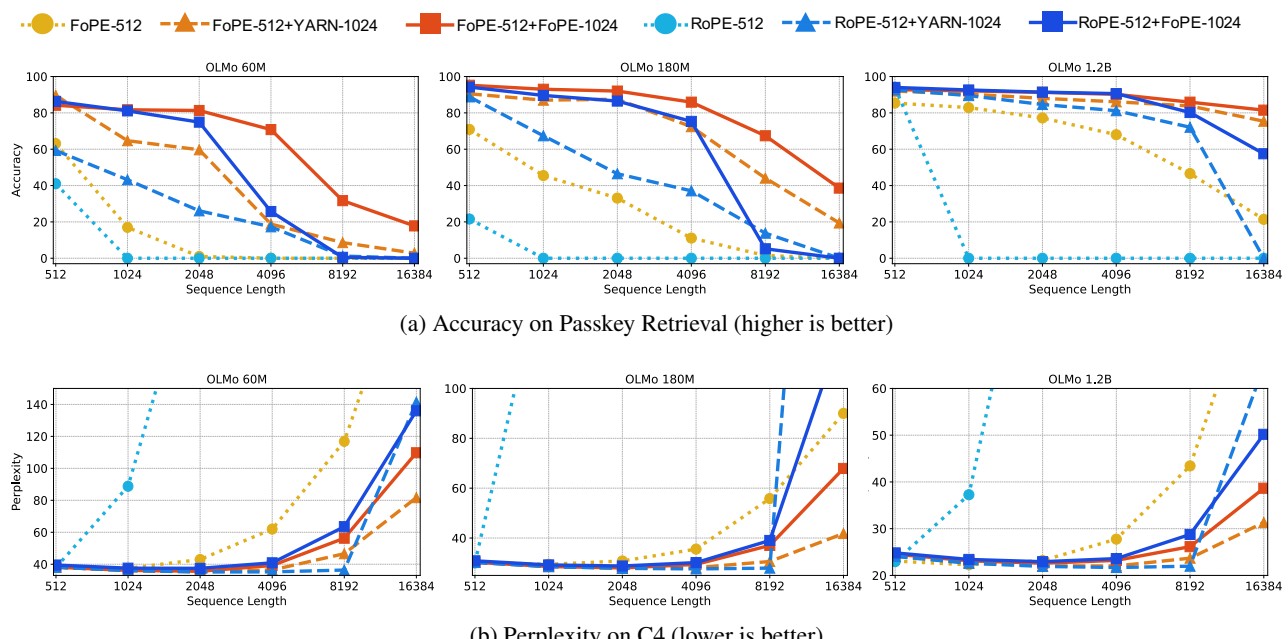

(a) Accuracy on Passkey Retrieval (higher is better)

(b) Perplexity on C4 (lower is better)

*Figure 4.* Effectiveness of FoPE in length extrapolation. Starting point models trained with 512 context length are extrapolated using YARN and FoPE on a corpus with a maximum sequence length of 1024. The generalization ability of FoPE not only surpasses that of RoPE as a base position embedding, but FoPE also demonstrate capability as an extrapolation method applied to pre-trained models.

# 4. Fourier Position Embedding

To mitigate the negative affect of the non-ideal frequency-domain properties in LMs, we propose ***Fourier Position Embedding (FoPE)*** to modify frequency-domain properties of attention:

**Treating Each Dimension as Multi-Frequency.** Although Linear Layers and Activation Functions bring serious Spectrum Leakage and Spectrum Distortion, they are crucial for enhancing expressive capacity. Therefore, we keep these modules unchanged but focus on modifying how attention processes information within each dimension.

To achieve this, we replace the single frequency in each dimension with Fourier Series:

$$h_m(n) = H_m(n)(e^{i\omega_m n} + \sum_\omega a_\omega e^{i\omega n}) \qquad (11)$$

where $a_\omega < 1$ because $\omega_m$ is the dominant frequency. This allows attention modules to capture multi-frequency information in each dimension.

We initialize vector $\{\omega_m\}$ as same as RoPE, and initialize vector $\{\omega\}$ and matrix $\{a_\omega\}$ based on the analysis in Sec 3.2: For $\{\omega\} \in \mathbb{R}^D$, we make sure $M \leq D$ so that $\{\omega_m\} \subseteq \{\omega\}$, and the other frequencies can be sampled within $[0, \pi]$ in any distribution. For $\{a_\omega\} \in \mathbb{R}^{D \times M}$, we initialize it with $N(0, \sigma)$ based on the hypothesis that the Spectrum Damage obeys the similar distribution as the Linear Layers. The coefficients for the real and imaginary part of the frequency

are sampled separately in our implementation, which can also use the same coefficient. The $D$ and $\sigma$ are kept as hyper-parameters to be adjusted.

**Zero-out Under-trained Frequencies.** As analyzed in Sec 3.3, the inadequate training of extremely-low frequencies $\omega_m < \frac{2\pi}{N}$ impairs the frequency-domain properties of attention. Thus, we define the floor frequency as $\omega_l = \frac{2\pi}{N}$, and clip the frequencies under the floor frequency to zero.

We choose zero as the substitute because the zero-frequency component can has the shortest and the longest wavelength at the same time, which presents both long-term and short-term dependency. Also, zero-frequency brings zero average position embedding and will not bring positional bias, which benifits length generalization.

**Overall function of FoPE** can be formalized as:

$$h_m(n) = H_m(n)f(\omega_m) \qquad (12)$$

$$f(\omega_m) = \begin{cases} 1 & , \omega_m < \omega_l \\ e^{i\omega_m n} + \sum_\omega a_\omega e^{i\omega n} & , \omega_m \geq \omega_l \end{cases} \qquad (13)$$

which treats each dimension either as a Fourier Series or as a zero-frequency component.

FoPE can be easily achieved with a weight matrix $W^F \in \mathbb{R}^{D \times (M - M_0)}$, where $M_0$ is the number of zero-frequency components in each head (details in Appendix B)

# 5. Experiments

To demonstrate the effectiveness of FoPE as both a position embedding and an extrapolation method, we conduct experiments during pre-training (Sec. 5.2), continual pre-training (Sec. 5.3) and fine-tuning (Sec. 5.4). Additionally, we perform ablations to analyze the impact of hyperparameters on FoPE (Sec. 5.5), and conduct empirical analysis to showcase the working machenism of FoPE (Sec. 5.6). Experimental details and results on more downstream tasks are shown in Appendix A.4.

## 5.1. Basic Settings

**Pre-training and Continual Pre-training:** In these two settings, we conduct experiments with the OLMo (Groeneveld et al., 2024) framework and consider different scale models having 60M, 180M, 1.2B parameters. We consider two metrics: perplexity for validation set and accuracy on Passkey Retrieval (Mohtashami & Jaggi, 2023). Passkey Retrieval measures the models' ability in retrieving a short passkey (i.e., a five-digit number) from a large context full of meaningless text. We conduct this evaluation based on the implementation from (Peng et al., 2023). During evaluation, the passkey is randomly positioned at uniformly distributed locations within the context. For each context length, we test for 1000 trials to ensure the positions sampled are sufficiently dispersed.

**Fine-tuning:** To assess the effectiveness of our method on more complex downstream tasks, we select SmolLM-1.7B (Allal et al., 2024), a capable open-source model with the same architecture as Llama (Touvron et al., 2023), as our base model. We then fine-tune it with the same training recipes as SmolLM-1.7B-Instruct (Allal et al., 2024). In this setting, we evaluate two types of downstream tasks, summarization and few-shot question-answering. For summarization, we use GovReport (Huang et al., 2021) and MultiNews (Fabbri et al., 2019). For few-shot question-answering, we use TREC (Li & Roth, 2002), TriviaQA (Joshi et al., 2017), and SAMSum (Gliwa et al., 2019). All these evaluations are conducted under the same setup as in (Bai et al., 2023).

## 5.2. Length Generalization after Pre-Training

We consider two settings to evaluate both the intra-domain and out-of-domain generalization:

**Setting 1:** We train models with a 10B-tokens subset of C4 (Raffel et al., 2020) and evaluate the perplexity in a validation set from C4.

**Setting 2:** We train models with ∼5B tokens from Gutenberg Books (Hart, 2007) and evaluate them in the same validation set as Setting 1. In this setting, the language distribution is different between the validation set and the

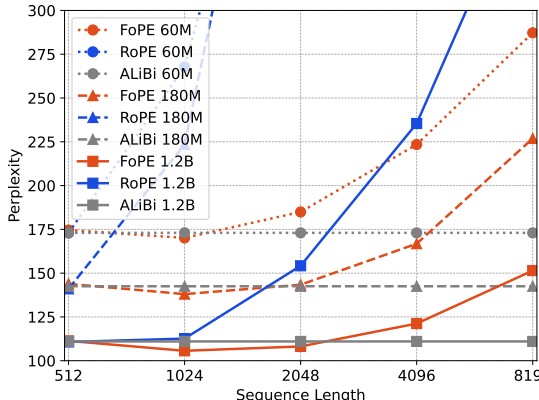

*Figure 5.* Training with max_seq_length=512 on Gutenberg Books and evaluating on a validation set of C4, FoPE also demonstrates its ability to generalize across different data distributions.

training set, which can further evaluate the generalization ability of different methods.

**Results of Perplexity.** (See Fig 1.b & 5) In both settings, FoPE shows a significant advantage over RoPE. But FoPE is slightly worse than ALiBi, as there is an issue when ALiBi meets this training corpus, which is also mentioned in other papers (Peng et al., 2023; Chen et al., 2024a). On the one hand, the corpus in C4 and Books mainly have short-distance dependency, thus the information from a short context window is enough for the prediction of almost all tokens. On the other hand, AliBi uses linear declined attention to eliminate long-distance information, and only pays attention to short-distance dependency. Based on these two reasons, ALiBi does not have any decline in perplexity as the context length increases.

**Results of Passkey.** (See Fig 1.a) FoPE demonstrates a significant advantage over all baselines in this task. RoPE's accuracy drops sharply to zero at twice the training length and remains at zero for longer sequences. ALiBi shows a linear decline in accuracy, further illustrating that its linearly declining attention is unable to capture information from long distances. In contrast, FoPE maintains stable retrieval accuracy at any position, demonstrating a strong ability to extract subtle information from long sequences.

## 5.3. Length Generalization after Continual Pre-Training

Beyond the use of positional embeddings during the original pre-training phase, several extrapolation methods (Peng et al., 2023; Chen et al., 2024a) have been proven critical for enhancing length generalization. Thus, we investigate two key aspects of FoPE: 1) whether existing extrapolation methods are also effective for FoPE; 2) whether FoPE can enable extrapolation on RoPE-based models, thereby allowing seamless integration with existing open-source models.

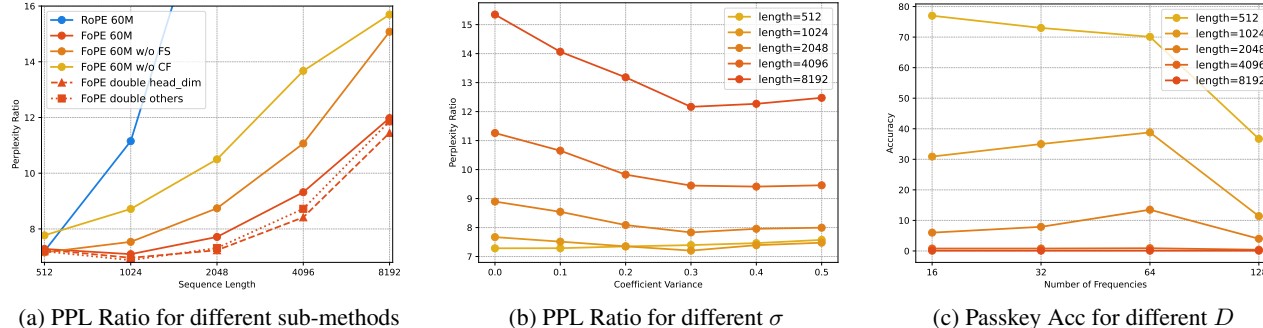

(a) PPL Ratio for different sub-methods     (b) PPL Ratio for different $\sigma$     (c) Passkey Acc for different $D$

*Figure 6.* Ablation studies on several hyperparameters. (a) and (b) evaluate PPL Ratio = $\text{PPL}_{c4}/\text{PPL}_{books}$, while (c) evaluates accuracy on Passkey Retrieval as we found $D$ slightly influence the perplexities of training. See the detailed analysis in Sec 5.5.

*Table 1.* Evaluation on Summarization and Few-shot QA tasks. On most benchmarks, FoPE delivers better length generalization compared to RoPE. The only exception is TriviaQA, where FoPE performs slightly worse in shorter contexts. However, FoPE remains stable up to 8k+ and outperforms RoPE significantly in 8k+.

| Benchmark | Length | RoPE | FoPE |
|---|---|---|---|
| GovReport | 0-4k | 13.02 | 13.27 ↑ 0.25 |
| | 4-8k | 11.35 | 12.50 ↑ 1.15 |
| | 8k+ | 12.02 | 12.38 ↑ 0.36 |
| MultiNews | 0-4k | 12.71 | 12.92 ↑ 0.21 |
| | 4-8k | 11.11 | 12.98 ↑ 1.87 |
| | 8k+ | 10.85 | 12.23 ↑ 1.38 |
| TREC | 0-4k | 37.00 | 41.00 ↑ 4.00 |
| | 4-8k | 42.00 | 56.00 ↑ 14.0 |
| | 8k+ | 36.00 | 51.00 ↑ 15.0 |
| TriviaQA | 0-4k | 36.50 | 33.26 ↓ 3.24 |
| | 4-8k | 36.12 | 33.53 ↓ 2.49 |
| | 8k+ | 25.02 | 33.87 ↑ 8.85 |
| SAMSum | 0-4k | 10.27 | 19.77 ↑ 9.50 |
| | 4-8k | 6.37 | 15.85 ↑ 9.48 |
| | 8k+ | 8.49 | 17.26 ↑ 8.87 |

In this sub-experiment, we select a representative extrapolation method, YARN (Peng et al., 2023), as our baseline. We fine-tune the last checkpoint from pre-training for $\sim$ 1B tokens in this setting.

**Results.** (See Fig 4) In comparison with RoPE+YARN, FoPE+YARN achieves significantly better length generalization performance, as demonstrated by lower perplexity on the C4 dataset and higher accuracy in the Passkey Retrieval task. Moreover, FoPE outperforms YARN in length extrapolation for both RoPE-based and FoPE-based models. These findings underscore the effectiveness and practical utility of FoPE, which holds the potential to enhance all RoPE-based open-source models.

### 5.4. Length Generalization after Fine-Tuning

We further evaluate the effectiveness of FoPE within the widely used LLaMA architecture, focusing on more challenging tasks such as summarization and few-shot question answering. For this purpose, we select SmolLM-1.7B, which is pre-trained with a context length of 2048. We then fine-tune SmolLM to a context length of 4096 using both RoPE and FoPE, and compare their performance on samples of varying sequence lengths from selected datasets, categorized into 0–4k, 4–8k, and 8k+.

**Results.** (See Table 1) On nearly all benchmarks, FoPE not only improves inner-domain performance for samples with 0–4k length, but also demonstrates more stable generalization capabilities for longer context lengths. The only exception occurs in TriviaQA, where FoPE performs slightly worse in shorter contexts. However, FoPE's performance remains stable up to 8k+ and significantly outperforms RoPE.

### 5.5. Ablation Studies

Considering the consistent performance of FoPE across different parameter scales and datasets, we conduct ablation studies only on the 60M models trained on 5B tokens from Gutenberg Books. Except for the primary ablation variable, all other hyper-parameters are kept identical to the main experimental settings.

**Both sub-methods of FoPE are useful** (See Fig 6a). FoPE is constitutive of two parts, called *Fourier Series (FS)* and *Clip Floor to Zero (CF)*. Although these two sub-methods are both useful for length generalization, combining them together brings a more significant improvement. On one hand, FS contributes more to length generalization, which demonstrates that the Spectrum Damage have a significant influence on length generalization. On the other hand, CF contributes more to fitting the current dataset and sequence length, which implies the zero-frequency component is the most informative and indispensable component.

**Increasing the dimension of attention heads is more beneficial than increasing the number of attention heads or layers** (See Fig 6a). More dimensions introduce more frequency components, making attention more robust to Spectral Damage. In contrast, adding more attention heads and layers aggravates Spectrum Damage, which diminishes the benefits of expanding the parameter scale.

**Variance $\sigma$ of $\{a_\omega\}$** (See Fig 6b). We keep $D = 16$ to only evaluate $\sigma$'s influence. By grid searching $\sigma$ from 0 to 0.5, we find that setting $\sigma = 0.3$ for 60M model obtain the best perplexity, especially for longer context. The best $\sigma$ implies the estimated strength of Spectrum Damage of the 60M model, and the estimation may become larger as the models' parameter scale increases.

**Number $D$ of $\{\omega\}$.** We keep $\sigma = 0.3$ to only evaluate $D$'s influence. By grid searching $\sigma$ from 16 to 128, we find that $D$ does not significantly influence the perplexity, but it is important for Passkey Retrieval. Setting $D = 64$ can obtain the best accuracy for Passkey Retrieval. The best $D$ is the estimated number of strong enough noisy components of each model, and this number may become larger as the parameter scale increases. The harmonic frequencies tend to be weaker than the base frequencies, and this phenomenon is more significant to the higher-order harmonics. Thus, there are limited noisy frequency components that have enough intensity to disturb the passing of the base wave, paying attention to not important components hinders the effectiveness of the model.

Although the choice of hyper-parameters affects the effectiveness of FoPE, the ablation study shows that FoPE consistently outperforms RoPE across all hyperparameter settings, demonstrating its robustness to hyperparameter selection.

*Table 2.* The loss of 20M toy models trained on a sequence length of 512. We consider three types of position embeddings, among which only RoPE has frequencies that cannot complete full cycles. For RoPE-A, all frequencies are adjusted from RoPE to the nearest values that exactly complete full cycles.

| Sequence Length | 512 | 1024 | 2048 | 4096 | 8192 |
|---|---|---|---|---|---|
| RoPE | 5.50 | 6.01 | 6.58 | 6.99 | 7.16 |
| RoPE + QK_Norm | 5.46 | 5.56 | 5.89 | 6.32 | 6.66 |
| RoPE-A | 5.72 | 5.86 | 6.18 | 6.46 | 6.67 |
| RoPE-A + QK_Norm | 5.69 | 5.89 | 6.27 | 6.59 | 6.81 |
| NoPE | 5.65 | 6.03 | 6.60 | 6.81 | 6.99 |
| NoPE + QK_Norm | 5.59 | 6.18 | 6.87 | 7.10 | 7.43 |

## 5.6. Empirically Validation of FoPE's Mechanism

To further validate the theoretical insights of FoPE analyzed before, we conduct empirical experiments on its two core modification of RoPE.

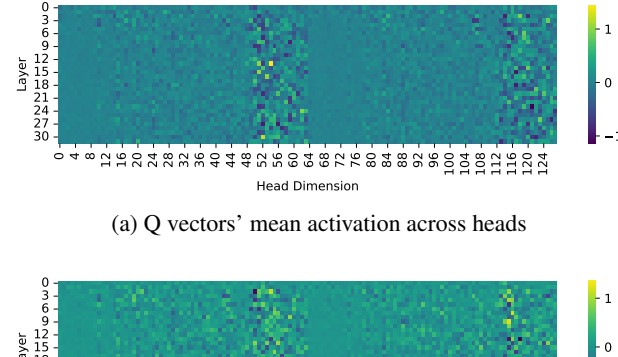

(a) Q vectors' mean activation across heads

(b) K vectors' mean activation across heads

*Figure 7.* The activation values across different dimensions of $q$, $k$ vectors in various layers, averaged over the attention heads in Llama2-7B. We found that under-trained dimensions holds higher absolute values across all layers of the model.

**Modeling each dimension as Fourier Series.** We use a toy model consisting of a single-layer MLP with an activation function, to validate the necessity of multi-frequency representation in a single dimension by simulating attention scores' calculation. For simplicity, we model the interaction of a token with only two dimensions (frequency components) hidden states, setting both Query and Key matrices as identity matrices during attention score computation. We track the transformations of each dimension through the MLP and activation function, matching frequency components with corresponding sinusoidal functions to obtain frequency-matched attention scores, termed Ground Truth (See Fig 3). Comparing Ground Truth with RoPE and FoPE attention scores, we find that FoPE more accurately captures multi-frequency periodicity, yielding attention scores that better align with information transfer.

**The negative effect of under-trained dimensions.** By visualizing the numerical expectations of the $q$ and $k$ vectors in each dimension (details in Appendix A.1), we observe that the absolute values of the dimensions corresponding to under-trained frequencies holds higher absolute values across all layers of the model. This positional bias may adversely affect robustness to out-of-domain rotation matrix values during length generalization. To further verify this hypothesis, we train several toy models (See Table 2). For models with QK_Norm, we normalize the $q$,$k$ vectors (enforcing a mean of 0 and variance of 1) before applying rotation matrix to eliminate positional bias. Such normalization delivers positive impacts only on under-trained frequency components, but not on frequency components complete full cycles. These experimental results validate our hypothesis.

## 6. Related Work

**Frequency-Domain Embedding.** Discrete Fourier Transform (DFT) (Oppenheim et al., 1982) has been widely used in various areas having periodic signals (Edfors et al., 2000; Sanchez, 2010). In machine learning, (Uteuliyeva et al., 2020; Lin et al., 2024b; Tancik et al., 2020; Tamkin et al., 2020; Lee-Thorp et al., 2022; Gillman et al., 2024) employed Fourier features into neural networks to enhance performance on NLP or CV tasks. S4 (Gu et al., 2021) also leveraged FFT and IFFT to shift its core computation into the frequency-domain, delivering more efficient computation. (Wang et al., 2019; Su et al., 2024) improved the attention mechanism by defining position embedding with complex number, while "phase" used in these methods is a typical concept in frequency-domain.

**Length Generalization.** Due to resource constraints, LMs are trained on limited-length corpus chunks and struggle with longer contexts (Voita et al., 2023; Dong et al., 2024; Hong et al., 2024; Qi et al., 2024). While absolute position embeddings (Vaswani, 2017) restrict the general use of positional information, methods as (Shaw et al., 2018; Yang, 2019) directly adjust the attention mechanism, another intuitive method is to redesign the position embedding (Press et al., 2021; Chi et al., 2022; Li et al., 2024; Kazemnejad et al., 2024; Su et al., 2024; Wang et al., 2024; Choromanski et al., 2024; Barbero et al., 2024; Chen et al., 2024b). Among these, RoPE (Su et al., 2024) encodes positional information using the phase of complex numbers, leveraging their periodicity to enhance access to long-distance dependencies. Several training-free or fine-tuning-based methods can also improve the LM's length generalization by refining RoPE (Peng et al., 2023; Chen et al., 2024a; Jin et al., 2024; Lin et al., 2024a). However, these works mainly address the drawbacks of RoPE in attention mechanism, neglecting the influence of other components in LMs.

## 7. Conclusion

In this paper, we analyze RoPE-based attention by modeling it in the frequency domain using *Discrete Signal Processing (DSP)* theory. Our analysis reveals that RoPE achieves periodic attention by implicitly performing *Non-Uniform Discrete Fourier Transform (NUDFT)*. However, this periodicity is corrupted by the non-ideal spectrum properties introduced by several parts in LMs. We propose *Fourier Position Embedding (FoPE)* to enhances attention's periodic extension and length generalization. FoPE models each dimension as Fourier Series and zero-out inadequately-trained frequency components. Experiments are conducted on models with various architectures and scales. Based on the evaluation on diverse well-known datasets including summarization and question-answering tasks, FoPE demonstrates significantly improvement length generalization com-

pared to baselines. Our ablation studies brings analyze on the influence of each sub-method and hyperparameter on FoPE. Several empirical toy experiments also provide further support for our method and theoretical modeling.

## Impact Statement

Our DSP-based modeling in the frequency domain provides a novel perspective for LMs to enhance length generalization and explore broader applications. These include aligning frequency-domain representations for better model collaboration, optimizing kv-cache compression via spectral analysis, and improving semantic communication with learnable frequency-domain embeddings.

## Acknowledgement

This work is supported by the National Science and Technology Major Project (2023ZD0121403), the Beijing Natural Science Foundation (IS23059), the Young Elite Scientists Sponsorship Program by CAST (2023QNRC001), and the National Natural Science Foundation of China (No. 62406165). We further extend our gratitude to Yihao Liu, Yiming Shi, Yizhou Jiang, Hang Deng and Chuxuan Shan, for their insightful discussion with us.

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

# A. More Experimental Results

## A.1. Visualization of $q$,$k$ vectors before applying RoPE

We conduct visualization experiments using the Llama2-7B model, which features an attention module with 32 heads, each comprising 128 dimensions, and a pretraining sequence length of 4096 tokens. The number of cycles sampled by each sinusoidal function during pretraining is calculated as $r_i = \frac{\theta_i L_{train}}{2\pi}$, where $i$ denotes the dimension index. Based on this, we determined that the dimensions corresponding to incomplete cycles fall within the ranges $[45, 64] \cup [109, 128]$.

We randomly sample 1000 tokens and compute the average activation values across heads for each dimension. The average activation values for every dimension of each layer are then plotted in Fig 7. The absolute activation values are significantly higher for dimensions corresponding to undertrained frequencies compared to others. This indicates that the RoPE rotation matrix pattern during pretraining has a notable impact on the distribution of $q$,$k$ vector activations.

## A.2. Influence of undertrained components in time-domain

We conduct visualization and ablations to further investigate the influence of undertrained components in RoPE.

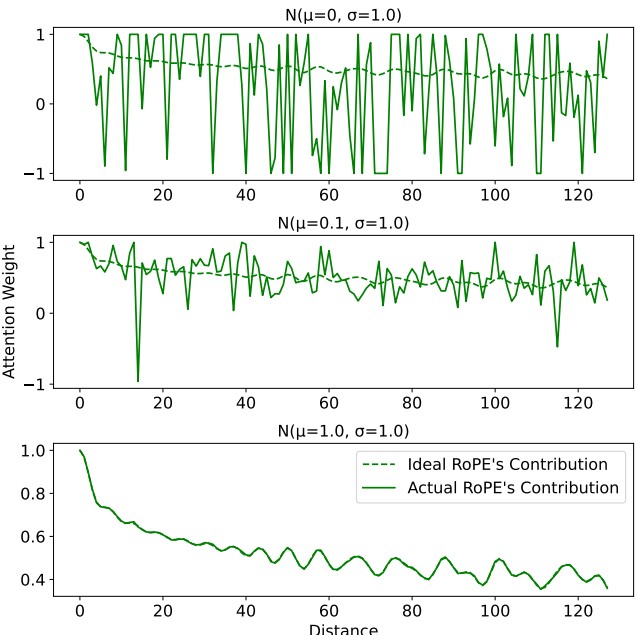

*Figure 8.* The statistical average contribution of RoPE to attention scores. We sample 1k Q and K vectors from Gaussian distributions with mean of 0, 0.1, and 1.0. The long-distance decay effect weakens as the mean decreases, disappearing entirely when the mean is 0.

In Fig 8, we visualize the time-domain pattern of RoPE. To get the "Ideal RoPE's Contribution", we suppose the Query and Key vectors are equal to 1 constantly (as in the original paper of RoPE (Su et al., 2024). To get the "Actual RoPE's Contribution", we suppose the Query and Key vectors obey the Gaussian distribution and sample 1000 times to get the mathematical expectation of the "Actual RoPE's Contribution". It can be seen that RoPE brings decay in attention score in its naive setting, which is brought by the undertrained components in RoPE. We hypothesis that this decay brings positional bias that may adversely affect robustness to out-of-domain rotation matrix values during length generalization.

As the positional bias can be eased if the mean of Query and Key vectors is set to zero, we conduct further ablation studies to normalize the Query and Key vectors before attention. Based on the results in 2, only if the position embedding contains components that cannot complete full cycle, the normalization brings better length generalization. Thus, these components are partially proved to deliver negative affect by positional bias. This experiment also demonstrate that the positional decay does not have beneficial influence on length generalization.

## A.3. Setups of main experiments

Our main experiments are conducted with 4 cards NVIDIA A6000 (maximum GPU memory=48GB).

**Pre-Training Settings.** The pre-training of 60M/180M/1.2B OLMo in 10B tokens lasts for 10/20/100 hours, respectively. The time-consuming has a linear relation with the number of tokens in other settings. For all model scales and experimental settings, we select 6e-4 as the learning rate and warm-up for 10000 steps with cosine scheduler. While the mini-batchsize on each device is different for each model, we accumulate gradients until the global batchsize reaches 1024 in all experiments.

**Fine-Tuning Settings.** Our fine-tuning procedure totally aligns with the official SmolLM-1.7B-Instruct recipe (Allal et al., 2024). We fine-tune SmolLM-1.7B with approximately 350k samples for one epoch, using the AdamW optimizer with a learning rate of 3e-4 and a cosine scheduler with a warmup ratio of 0.1. The entire training process takes 4 hours on our machines.

*Table 3.* The hyper-parameter of different model scales. The upper part is the parameter shared by all experiments, while the lower part is the parameter specific to FoPE. We use different mini-batch sizes depending on the max_seq_length: the first corresponds to max_seq_length=512, and the second corresponds to max_seq_length=1024.

| Model Scale | 60M | 180M | 1.2B |
|---|---|---|---|
| **Num_Heads** | 8 | 8 | 16 |
| **Num_Layers** | 8 | 8 | 16 |
| **MLP_Ratio** | 8 | 8 | 8 |
| **Head_Dim** | 64 | 128 | 128 |
| **Mini_Batchsize** | 64/32 | 32/16 | 8/4 |
| **Var_Freq** $\sigma$ | 0.3 | 0.4 | 0.6 |
| **Num_Freq** $D$ | 64 | 128 | 128 |

**Evaluation Settings.** For pre-training, evaluations are conducted on the checkpoint from the last step. For fine-tuning, we save checkpoints every 100 steps and report the best result for each method. This is partly due to YARN (Peng et al., 2023) being prone to overfitting with excessive fine-tuning steps, a limitation not observed in FoPE.

## A.4. Experimental results on more benchmarks

In addition to evaluating the long-context capabilities of FoPE, we also assessed its performance on a variety of other tasks (see Tables 4, 5, and 6). The results show that FoPE performs comparably to, or even better than, other baselines, demonstrating that FoPE can enhance long-context capabilities without compromising performance on other tasks.

*Table 4.* Accuracy of 1.2B models with different positional embeddings on several downstream tasks.

| Method | basic arithmetic | social iqa | winogrande | openbook qa | sciq | hellaswag | piqa | commonsense qa | arc easy | avg. |
|---|---|---|---|---|---|---|---|---|---|---|
| NoPE | 25.67 | 43.71 | 51.86 | 29.80 | 76.70 | 41.83 | 68.83 | 31.61 | 51.40 | 42.14 |
| ALiBi | 24.97 | 42.53 | 53.12 | 31.40 | 77.70 | 43.06 | 69.42 | **33.42** | **53.68** | 42.93 |
| KERPLE | 25.03 | 43.81 | **54.07** | 32.40 | **78.20** | 43.65 | 69.64 | 32.92 | 52.46 | 43.22 |
| FIRE | 25.60 | 42.63 | 49.88 | 33.40 | 77.00 | 42.75 | 69.31 | 32.92 | 50.35 | 42.38 |
| RoPE | 24.60 | 43.45 | 51.54 | **33.60** | 77.10 | 43.36 | **70.13** | 33.01 | 52.98 | 42.98 |
| FoPE | **26.17** | **44.12** | 53.20 | 32.20 | 77.80 | **43.83** | 70.08 | 32.92 | 53.33 | **43.37** |

*Table 5.* Cross-Entropy Loss of 1.2B models with different positional embeddings on several downstream tasks.

| Method | natural qs open | trivia qa wiki | arc easy | avg. |
|---|---|---|---|---|
| NoPE | 1.4334 | 1.6129 | 1.3077 | 1.4513 |
| ALiBi | 1.3879 | 1.6057 | 1.2789 | 1.4242 |
| KERPLE | 1.4149 | 1.5878 | 1.2825 | 1.4284 |
| FIRE | 1.4365 | 1.6258 | 1.3118 | 1.4580 |
| RoPE | 1.4114 | 1.5973 | 1.2588 | 1.4225 |
| FoPE | **1.3818** | **1.5736** | **1.2272** | **1.3941** |

Table 6. Evaluation of 1.2B models with different positional embeddings on mmlu.

| Method | stem | social science | humanity | other | avg. |
|---|---|---|---|---|---|
| NoPE | 25.47 | 25.38 | 28.01 | 24.39 | 25.81 |
| ALiBi | 25.93 | 27.20 | 26.44 | 24.39 | 25.99 |
| KERPLE | 26.02 | 25.97 | 27.82 | 24.82 | 26.16 |
| FIRE | 26.34 | **27.41** | 26.25 | 24.16 | 26.04 |
| RoPE | 27.01 | 27.26 | 27.87 | 24.53 | 26.68 |
| FoPE | **27.30** | 27.31 | **29.89** | **25.79** | **27.57** |

## B. Implementation Details

**Implementation of FoPE** can be easily achieved with a weight matrix $W^F \in \mathbb{R}^{D \times (M - M_0)}$, where $M_0$ is the number of zero-frequency components in each head. This matrix maps the coefficients of all frequencies to a Fourier Series for each dimension. Since the zero-frequency sinusoidal function does not affect the original hidden states, the output dimension is less than the dimension of each head. To introduce more diversity and better simulate the randomness of the Spectrum Damage, we assign separate weights for different heads, as well as for the cosine and sine functions. In our implementation, gradients are not required for these matrices, so FoPE adds negligible memory and computation overhead compared to RoPE.

**Pseudo-code** of FoPE is shown in the final pages.

## C. Theoretical Details

### C.1. Derivation of Lemma 3.1

Given a non-linear function g(x), it can be rewritten as a power series by Taylor expansion:

$$g(x) = \sum_{p \in \mathbb{N}} a_p x^p \tag{14}$$

Suppose the input is a double-frequencies function:

$$x(n) = \cos \omega_1 n + \cos \omega_2 n \tag{15}$$

, the output becomes:

$$g(x(n)) = \sum_{p \in \mathbb{N}} a_p (\cos \omega_1 n + \cos \omega_2 n)^p \tag{16}$$

Considering the product conversion formula of sinusoid function:

$$\cos \alpha \cos \theta = \frac{1}{2}[\cos(\alpha - \theta) + \cos(\alpha + \theta)] \tag{17}$$

Thus, each sub-function of Eq.(16) can generate harmonic functions. For example, when $p = 2$:

$$
\begin{aligned}
(\cos \omega_1 n + \cos \omega_2 n)^2 &= (\cos \omega_1 n)^2 + (\cos \omega_2 n)^2 + 2\cos \omega_1 n \cos \omega_2 n \\
&= 1 + \frac{1}{2}\cos 2\omega_1 n + \frac{1}{2}\cos 2\omega_2 n + \cos(\omega_1 - \omega_2)n + \cos(\omega_1 + \omega_2)n
\end{aligned}
\tag{18}
$$

### C.2. Derivation of Equal (10)

Before we begin the derivation, let's familiarize with two functions in time domain: **Rectangular Pulse Function** and **Single-Frequency Function**.

Given a **Rectangular Pulse** function:

$$r(t) = \begin{cases} 1, & |t| \le T \\ 0, & |t| > T \end{cases} \tag{19}$$

whose Fourier Transform is a **Sa Function**:

$$R(\omega) = \int_{-T}^{T} e^{-j\omega t} dt = 2\frac{\sin(\omega T)}{\omega} \tag{20}$$

Given a **Single-Frequency** function:

$$f(t) = e^{j\omega_m t} \tag{21}$$

whose Fourier Transform is:

$$F(\omega) = \int_{-\infty}^{+\infty} e^{j\omega_m t} e^{-j\omega t} dt \tag{22}$$

which is equal to an **Impulse Function** in frequency domain:

$$\delta(\omega_m) = \begin{cases} 1, & \omega = \omega_m \\ 0, & \omega \neq \omega_m \end{cases} \tag{23}$$

Then, Eq (10) can be easily derivated, as the context $x$ truncated with length $T$ can be seen as:

$$x(t) = f(t) \cdot r(t) \tag{24}$$

Its Fourier Transform in discrete case is Eq. (10).

### C.3. Low-Pass Property of RoPE

As the frequency of RoPE obey $\omega_m = 1/\theta^{(2m/M)}$, it only samples frequency less than 1 and samples more densely near 0.

According to the following Theorems and Corollary, RoPE only filters the low-frequency components from hidden states for attention, thus achieving a low-pass filter in attention.

**Theorem C.1.** *For any continuous function, to reconstruct the original function using its discrete samples without distortion, the relation between the sampling frequency $f_s$ and the highest frequency component $f_h$ of the function must obey: $f_s \geq 2f_h$.*

**Theorem C.2.** *For any discrete function, the sampling frequency $\omega_s$ is constant: $\omega_s = 2\pi$.*

**Corollary C.3.** *The frequency $\omega$ of any discrete function obey: $\omega \leq \pi$.*

Comparing with high band $\omega \in (\pi/2, \pi]$, lower frequency components delivers a longer term influence on other tokens in time domain, letting attention have stronger bias to grab long-distance information.

### C.4. The Relationship Between the Theoretical Modeling and Practical Implementation of RoPE

Let's define the query and key in the complex domain as $\mathbf{q} = ||q||e^{i\theta_q}$ and $\mathbf{k} = ||k||e^{i\theta_k}$, in the matrix domain as $\vec{\mathbf{q}} = [q_x, q_y]$ and $\vec{\mathbf{k}} = [k_x, k_y]$. According to the geometrical meaning of complex number, the relationships between the variations above are $||q|| = \sqrt{q_x^2 + q_y^2}$ and $\theta_q = \arctan\frac{q_x}{q_y} + k\pi$, with similar expressions holding for $||k||$ and $\theta_k$.

Thus, the implementation based on the rotary matrix reflects the geometric interpretation of phase rotation in the complex domain, as illustrated by the following derivation:

$$\langle f_q(\mathbf{x_m}, m), f_k(\mathbf{x_n}, n) \rangle \tag{25}$$

$$= \langle \mathbf{q}e^{im\theta}, \mathbf{k}e^{in\theta} \rangle \tag{26}$$

$$= Re[\mathbf{q}\mathbf{k}^* e^{i(m-n)\theta}] \tag{27}$$

$$= Re[\|q\|\|k\| e^{i[\theta_q - \theta_k + (m-n)\theta)]}] \tag{28}$$

$$= \|q\|\|k\| \cos(\theta_q - \theta_k + (m-n)\theta) \tag{29}$$

$$= \|q\|\|k\| [\cos(\theta_q - \theta_k)\cos(m-n)\theta - \sin(\theta_q - \theta_k)\sin(m-n)\theta] \tag{30}$$

$$= \|q\|\|k\| [(\cos\theta_q \cos\theta_k + \sin\theta_q \sin\theta_k)\cos(m-n)\theta - (\sin\theta_q \cos\theta_k - \cos\theta_q \sin\theta_k)\sin(m-n)\theta] \tag{31}$$

$$= (q_x k_x + q_y k_y)\cos(m-n)\theta - (q_y k_x - q_x k_y)\sin(m-n)\theta \tag{32}$$

$$= \begin{bmatrix} q_x & q_y \end{bmatrix} \begin{bmatrix} \cos(m-n)\theta & -\sin(m-n)\theta \\ \sin(m-n)\theta & \cos(m-n)\theta \end{bmatrix} \begin{bmatrix} k_x \\ k_y \end{bmatrix} \tag{33}$$

$$= \begin{bmatrix} q_x & q_y \end{bmatrix} \begin{bmatrix} \cos m\theta & \sin m\theta \\ -\sin m\theta & \cos m\theta \end{bmatrix} \begin{bmatrix} \cos n\theta & -\sin n\theta \\ \sin n\theta & \cos n\theta \end{bmatrix} \begin{bmatrix} k_x \\ k_y \end{bmatrix} \tag{34}$$

$$= (\mathbf{R}_{\theta,m} \vec{\mathbf{q}})^T (\mathbf{R}_{\theta,n} \vec{\mathbf{k}}) \tag{35}$$

$$= \vec{\mathbf{q}}^T \mathbf{R}_{\theta,m-n} \vec{\mathbf{k}} \tag{36}$$

```
1  class FourierEmbedding(RotaryEmbedding):
2      def __init__(self, config):
3          super().__init__(config)
4
5          self.input_dim = self.inv_freq.size(-1)
6          self.output_dim = self.input_dim if self.input_dim <= self.head_dim//4 else self.
               head_dim//4
7
8          self.sin_coef = nn.Parameter(
9              torch.randn(self.config.n_heads, self.input_dim, self.output_dim),
10             requires_grad=False
11         )
12         self.cos_coef = nn.Parameter(
13             torch.randn(self.config.n_heads, self.input_dim, self.output_dim),
14             requires_grad=False
15         )
16         torch.nn.init.xavier_normal_(self.sin_coef, gain=self.config.
               rope_fourier_init_norm_gain)
17         torch.nn.init.xavier_normal_(self.cos_coef, gain=self.config.
               rope_fourier_init_norm_gain)
18
19         self.sin_coef += torch.eye(self.input_dim, device=self.sin_coef.device)
20         self.cos_coef += torch.eye(self.input_dim, device=self.cos_coef.device)
21
22     def apply_rotary_pos_embed(self, pos_sin, pos_cos, t):
23         fourier_sin = torch.einsum("bhtD,_hDd_->_bhtd", pos_sin, self.sin_coef / self.
               sin_coef.sum(dim=-2, keepdim=True))
24         fourier_cos = torch.einsum("bhtD,_hDd_->_bhtd", pos_cos, self.cos_coef / self.
               cos_coef.sum(dim=-2, keepdim=True))
25
26         fourier_sin = F.pad(input=fourier_sin, pad=(0, self.head_dim//2-fourier_sin.size
               (-1)), mode="constant", value=1)
27         fourier_cos = F.pad(input=fourier_cos, pad=(0, self.head_dim//2-fourier_cos.size
               (-1)), mode="constant", value=1)
28
29         fourier_sin = torch.cat((fourier_sin, fourier_sin), dim=-1)
30         fourier_cos = torch.cat((fourier_cos, fourier_cos), dim=-1)
31
32         return ((t * fourier_cos) - (self.rotate_half(t) * fourier_sin)).to(t.dtype)
33
34 class RotaryEmbedding(nn.Module):
35     def __init__(self, config):
36         super().__init__()
37         self.config = config
38
39         self.dim = self.config.d_model // self.config.n_heads
40         self.inv_freq = self.get_inv_freq(self.dim)
41
42     def get_inv_freq(self, dim):
43         inv_freq = 1.0 / (
44                 self.config.rope_theta ** (torch.arange(0, dim, 2, device=device, dtype=
                       torch.float) / dim)
45         )
46         if self.config.fope is True:
47             inv_freq[inv_freq < 2*torch.pi/self.config.max_sequence_length] = 0
48             inv_freq = inv_freq[inv_freq != 0.0]
49
50         return inv_freq
```

```
1   def get_rotary_embedding(self, seq_len):
2       with torch.autocast(device.type, enabled=False):
3           seq = torch.arange(seq_len, device=device, dtype=torch.float)
4           freqs = torch.einsum("t, hd ->htd", seq, self.inv_freq)
5
6           if self.config.fope is True:
7               positions = freqs.unsqueeze(0)
8           else:
9               positions = torch.cat((freqs, freqs), dim=-1).unsqueeze(0)
10
11      return positions.sin(), positions.cos()
12
13  def rotate_half(self, x):
14      B, nh, T, hs = x.size()
15      x = x.view(B, nh, T, 2, hs // 2)
16
17      x1, x2 = x.unbind(dim=-2)
18
19      return torch.cat((-x2, x1), dim=-1)
20
21  def apply_rotary_pos_emb(self, pos_sin, pos_cos, t):
22      return ((t * pos_cos) - (self.rotate_half(t) * pos_sin)).to(t.dtype)
23
24  def forward(self, x, all_len):
25      with torch.autocast(x.device.type, enabled=False):
26          x_len = x_.shape[-2]
27          pos_sin, pos_cos = self.get_rotary_embedding(all_len)
28          pos_sin = pos_sin.type_as(x_)
29          pos_cos = pos_cos.type_as(x_)
30
31          x_ = self.apply_rotary_pos_emb(
32              pos_sin[:, :, all_len - x_len : all_len, :],
33              pos_cos[:, :, all_len - x_len : all_len, :],
34              x_,
35          )
36
37      return x_.type_as(x)
```

