# OpenReview forum: "Fourier Position Embedding: Enhancing Attention’s Periodic Extension for Length Generalization"
_ICML.cc/2025/Conference — ICML 2025 poster_

### Official Review · Reviewer_FbaF · 2025-03-09

**Overall Recommendation:** 3

**Summary:**

The paper analyzes how RoPE enables periodic attention patterns and then analyzes the limitation of RoPE in that regard.
The authors argue that the limitation arises from spectral damage prevalent when RoPE is used with typical DL architectures.
The authors propose FoPE, which is based on RoPE, but while RoPE treats each dimension as a single-frequency function, FoPE models each dimension as a Fourier Series, consisting of a dominant frequency component and several harmonic components.
Additionally, FoPE clips low frequencies because the authors argue these are undertrained.
In experiments, the authors show that FoPE maintains a better performance for increased context lengths.

**Claims And Evidence:**

The primary claim—that FoPE improves length extrapolation—is well-supported by empirical analysis. The experiments effectively showcase FoPE’s advantages over RoPE, and the inclusion of comparisons with YARN strengthens the argument.

**Essential References Not Discussed:**

I am not aware of any essential references that are missing.

**Experimental Designs Or Analyses:**

The experimental design seems to be valid and sound. Models up to 1.2B parameters are tested, which is already a reasonable size to support generalization to practical settings. The comparison optionally includes YARN, which strengthens the argument.
As I am not deeply familiar with related approaches, I don't know if a comparison to additional methods would be appropriate.
A comparison to non-transformer architectures that provide favourable length extrapolation capabilities would be interesting in addition (e.g. see Figure~7 in [1]).
That said I have one concern regarding the hyperparameters D and sigma. The authors rightly conduct a sensitive analysis which shows that the parameters seem to be important. However, there is no clear guidance on how to select this parameter a-priori.

[1] xlstm: Extended long short-term memory, M Beck et al. Advances in Neural Information Processing Systems 37

**Methods And Evaluation Criteria:**

The evaluation methods and criteria appear appropriate. The experiments test models up to 1.2B parameters, a reasonable scale for assessing generalization. The comparison with YARN adds credibility.

**Other Comments Or Suggestions:**

-

**Other Strengths And Weaknesses:**

Strengths:
- Clear motivation with according analysis of the approach (spectrum damage)
- Evalution includes comparision to length extrapolation technique (yarn)

Weakness:
- No guidance for hyperparameter selection (D and sigma) although the methods seem to be sensitive to these parameters.
- No comparison to non-transformer LLM architectures that provide more favorable length generalization properties like RWKV, xLSTM or Mamba (e.g. see Figure~7 in [1])

[1] xlstm: Extended long short-term memory, M Beck et al. Advances in Neural Information Processing Systems 37

**Questions For Authors:**

- How would you select the sigma and D hyperparameter for a new large-scale models where a hyperparameter search is not feasible because of the scale/cost of the model training? (The sensitivity of these parameters could affect the real-world usability of FoPE)
- Could clipping low frequencies have unintended consequences in certain applications?

**Relation To Broader Scientific Literature:**

The paper is closely related to research on positional embeddings for transformers and the broader literature on length extrapolation. The study builds on RoPE, a widely used technique in modern deep learning models.

**Theoretical Claims:**

I briefly reviewed the theoretical claims presented in the main paper and did not identify any obvious issues.

---

> ### Author Rebuttal · Authors · 2025-03-29
>
> Thanks for the valuable comments from the reviewer, we want to deliver further explanations accordingly.
> # Comparison with more baselines on more benchmarks
> It is a nice suggestion to compare FoPE with more baseline methods, thus we supplement the following experiments:
> - We validate the effectiveness of FoPE by evaluating it on 10+ additional benchmarks and comparing it with 3 more baselines. These supplementary experiments are conducted on OLMo-1.2B training on C4 datasets, with all experimental settings kept consistent with Sec 5.2 (line 293-329).
>
> |methods|avg acc|basic_arithmetic|social_iqa|winogrande|openbook_qa|sciq|hellaswag|piqa|commonsense_qa|arc_easy|
> |-|-|-|-|-|-|-|-|-|-|-|
> |NoPE|42.14|25.67|43.71|51.86|29.80|76.70|41.83|68.83|31.61|51.40|
> |ALiBi|42.93|24.97|42.53|53.12|31.40|77.70|43.06|69.42|**33.42**|**53.68**|
> |KERPLE|43.22|25.03|43.81|**54.07**|32.40|**78.20**|43.65|69.64|32.92|52.46|
> |FIRE|42.38|25.60|42.63|49.88|33.40|77.00|42.75|69.31|32.92|50.35|
> |RoPE|42.98|24.60|43.45|51.54|**33.60**|77.10|43.36|**70.13**|33.01|52.98|
> |FoPE|**43.37**|**26.17**|**44.12**|53.20|32.20|77.80|**43.83**|70.08|32.92|53.33|
>
> |methods|avg acc|mmlu_stem|mmlu_social_sciences|mmlu_humanities|mmlu_other|
> |-|-|-|-|-|-|
> |NoPE|25.81|25.47|25.38|28.01|24.39|
> |ALiBi|25.99|25.93|27.20|26.44|24.39|
> |KERPLE|26.16|26.02|25.97|27.82|24.82|
> |FIRE|26.04|26.34|**27.41**|26.25|24.16|
> |RoPE|26.68|27.01|27.26|27.87|24.53|
> |FoPE|**27.57**|**27.30**|27.31|**29.89**|**25.79**|
>
> |methods|avg ce loss|natural_qs_open|trivia_qa_wiki|arc_easy|
> |-|-|-|-|-|
> |NoPE|1.4513|1.4334|1.6129|1.3077|
> |ALiBi|1.4242|1.3879|1.6057|1.2789|
> |KERPLE|1.4284|1.4149|1.5878|1.2825|
> |FIRE|1.4580|1.4365|1.6258|1.3118|
> |RoPE|1.4225|1.4114|1.5973|1.2588|
> |FoPE|**1.3941**|**1.3818**|**1.5736**|**1.2272**|
>
> - **While FoPE is primarily designed for length generalization and long-context tasks, it also performs comparably or even surpasses all baseline methods**. But on MMLU, all methods have weak performance, perhaps caused by the training data. In the future work, we may conduct more experiments on datasets containing math data.
> - We also conduct evaluation similar as Sec 5.2 (See https://anonymous.4open.science/r/FoPE-Supplementry-Material/Supplementary_Main_Experiments.pdf). **FoPE still demonstrate the best accuracy on passkey retrieval  and the second-best ppl on C4 (only slightly behind ALiBi).**
> - As time is limited, we have not conducted experiments on RWKV, xLSTM and Mamba. But it is really a nice suggestion to consider the length generalization capabilities of models with different architectures. We would like to cite these papers in related work in our future revisions.
>
> # Clarification of the hyper-parameters' influence on FoPE
> We agree with the reviewer that hyper-parameters optimization is crucial for the large-scale training, but we would like to make some clarifications:
> 1.**In a wide range of parameters we tested, FoPE consistently demonstrates a significantly better performance than RoPE**.
> 2. **The performance of FoPE is not highly sensitive to hyper-parameters** (see ablation in Fig 6 and Sec 5.5), although an elaborate hyper-parameter selection of FoPE may lead to a better performance.
>
> Thus, **even without a careful selection of hyper-parameters, FoPE is still competent for replacing RoPE in Transformers**.
>
> Additionally, we emphasize that hyper-parameter tuning is an inevitable procedure before pre-training nowadays, many other hyper-parameters also need to be selected (i.e. hidden_dim, num_heads, ...). As for FoPE, the ablation studies in Fig 6 and Sec 5.5 suggested that:
> - The optimal $\sigma$ for FoPE increases as hidden_dim and num_layers grow, which is expected, as larger models suffer more from spectral damage.
> - The best $D$ tends to be slightly bigger than the head_dim of each attention head. This is because too few frequency components cannot represent the spectrum damage well, , while too many exceed the model’s representational capacity.
>
> But it is a nice suggestion to have a better selection strategy, we would like to continually research this problem.
>
> # Influence of clipping low frequencies
> It is an insightful suggestion to consider the negative influence of clipping low frequencies, we want to make some clarification below:
> - "Clipping low frequencies" is quite similar to the "interpolation" used by many extrapolation methods like YARN[1]. As this operation is mainstream solution for LLMs' length generalization, we suppose "Clipping low frequencies" does not have significant negative influence in most tasks.
> - **Also, FoPE has a consistent performance on diverse benchmarks, .** Thus, the answer to this problem is still unclear. But we would like to continually consider this issue.
>
> ---
>
> [1] YaRN: Efficient Context Window Extension of Large Language Models. ICLR 2024.

---

### Official Review · Reviewer_LcUb · 2025-03-15

**Overall Recommendation:** 3

**Summary:**

The paper introduces the Fourier positional embedding (FoPE) based on Fourier series. The authors begin by analyzing the rotary positional embedding (RoPE) method in the frequency domain. Further analysis of the feed-forward network yields additional information that linear layers produce spectrum leakage (mix of frequencies) and the activation functions lead to spectrum distortion due to their harmonics generation. These two types of spectrum effects are causes for spectrum damage for context length generalization, hindering RoPE’s effectiveness. Then, the attention mechanism is shown to produce spectrum damage on low frequencies for *undertrained* components. Following this analysis, FoPE is proposed as a multi-frequency representation for each dimension based on Fourier Series, instead of a single frequency like RoPE. The proposed method also includes zeroing the low frequency components to control for undertrained frequencies. Next the authors compare FoPE against RoPE and ALiBi on pre-training, and against YARN extension on continual pre-training of OLMo, and against RoPE on fine-tuning SmolLM-1.7B. They measure perplexity and passkey retrieval to test the context length generalization of the resulting models. In most cases, FoPE shows better generalization when increasing the context length at test time. Further, ablation analysis is performed to test both the zeroing of frequency and without the Fourier series, and various analysis related to other aspects of the language model.

## update after rebuttal
Thanks to the authors for clarifying the various points.  After a thoughtful consideration, primarily given the additional comparisons, I've decided to update my score.

**Claims And Evidence:**

The claims made in the submission are mostly supported. A formal definition of *undertrained* components of attention is missing, so it is unclear when does spectrum damage based on this definition matters. An empirical analysis of the potential Spectrum Damage on a well-trained model is lacking.

**Essential References Not Discussed:**

Not aware of such references.

**Experimental Designs Or Analyses:**

The proposed method is evaluated on a pre-training on C4 and Books datasets. The results show that ALiBi performs better than FoPE. In the text, the authors suggest that there is an “issue” with how ALiBi considers the linear declined attention, putting a considerable effort to learn the short-distance information. However, it is well-known that the distribution of mutual information between tokens behaves as a power-law distribution [1]. Namely, this is a property of the data, and the model exploits such prior knowledge (and not the other way around). In addition, pre-trained language models have better perplexity than the ones showed in Section 5.2, and the models are much bigger in size today (using more train tokens) than those used in the paper.

The passkey retrieval results in section 5.2 show that the bigger the model, the better the retrieval results. If the 3 FoPE model sizes have the same embeddings and trained with the same context length, then why are the smaller models unable to generalize in the same way as the biggest model? Their curves decay much faster. Also, existing models can solve tasks beyond 8k context length today. It is unclear if they can generalize similarly as FoPE or even better. This suggests that there may be an issue when training the models or with the experiment itself.

Last, it is unclear what is the maximum sequence length for section 5.4? Also, what is the performance of the baseline model *before* fine-tuning?

Other positional embedding methods have not been compared.

[1] “Critical Behavior in Physics and Probabilistic Formal Languages”, Lin and Tegmark, 2017

**Methods And Evaluation Criteria:**

The derived methods seem sound. The selection of the low frequency threshold is not analyzed. What is the impact on the method if an incorrect or poor threshold selected? The evaluation criteria for the method seems correct.  The datasets could be selected to test generalization further (e.g., perplexity on SCROLLS).

**Other Comments Or Suggestions:**

The descriptions in the text of the experiments related to Figure 6 look disconnected to the Figure itself.

**Other Strengths And Weaknesses:**

* Theory and derivations are valuable in the paper.
* The experiments and results are not conclusive. It’s hard to really conclude the value of FoPE versus RoPE or other not-compared embedding methods.

**Questions For Authors:**

See questions in sections above.
* Can you learn the threshold value for the low frequencies?

**Relation To Broader Scientific Literature:**

The contribution of FoPE and proposes that context length generalization is of some value to the community. Current LLMs have context lengths that are very large (100k+ tokens), thus the significance is low. The theoretical analysis is well executed but limited (i.e., how do residual connections influence the frequencies? How does the LayerNorm or the specific activation functions?).

**Theoretical Claims:**

The derivations in the paper seem correct. Please, see comments above about undertrained components.

---

> ### Author Rebuttal · Authors · 2025-03-30
>
> Thanks for the elaborate comments, we will address them according to their proposed order.
>
> # Clarification for "undertrained components" and the impact of poor threshold
> - **We have jointly defined the "undertrained components" and "floor frequency" in Fig 2, Sec 3.3 and Sec 4, using both visualization and formula.**
> - Also, **these two concepts are well-known in the area of length generalization**, mentioned by many papers [1, 2].
> - As for the impact of poor threshold, we add ablations on 60M OLMo trained with 512 context length (similar setting as Sec 5.2). **Our findings suggest that selecting a floor frequency no-less than $2\pi/L$ is necessary, but a higher threshold does not have a significant influence.**
> |threshold (f=$2\pi/L$)|0|0.5f|0.75f|f|1.25f|1.5f|
> |-|-|-|-|-|-|-|
> |**ce loss on 8192 length**|6.86|6.54|5.97|5.84|5.86|5.85|
>
> # Empirical analysis of the Spectrum Damage on well-trained models
> - In Sec 5.4, our experiments are based on SmolLM-1.7B, a famous model well-trained by HuggingFace for academic purposes.
> - In Sec 5.6, Fig 3 and Fig 7, we have a detailed empirical analysis of the influence of Spectrum Damage on LLaMA-2-7B.
> - To further illustrate this phenomenon, we conduct another empirical analysis on LLaMA-2-7B (See https://anonymous.4open.science/r/FoPE-Supplementry-Material/Empirical_Analysis_on_Spectrum_Damage.pdf). We force every token only has one frequency component in the 1st layer. Then, we compare the spectrum of the 1st and 2nd layer after DFT on the attention map. **Many other frequency components appear in the 2nd layer, which demonstrates the happening of Spectrum Damage.**
>
> # Advantage of FoPE over ALiBi and the well-known negative influence of linear declined attention
> - **Our results in Sec 5.2 clearly show FoPE's advantages compared to ALiBi, it is unclear why the reviewer got the opposite conclusion.**
> - We agree that "the mutual information between tokens behaves as a power-law distribution", and we appreciate this perspective.
> - However, this property is only useful for short context modeling, but not for long-context modeling. For example, **the "linear declined attention" hinders the retrieval of long-distance information, thereby lacking the crucial ability for long-context modeling.**
> - **Not only do our results in Fig 1 demonstrate this phenomenon, but many well-known work also have the similar viewpoint [2, 3, 4]** that "linear declined attention" and ALiBi is fall short in long-context applications.
>
> # Clarification for Sec 5.4 (the maximum sequence length and base model performance)
> - We have provided the maximum sequence length in Sec 5.4 (line 328 and 373).
> - The performance of SmolLM-1.7B-base is as follows, which do not influence our conclusions:
> |length|gov_report|multi_news|trec|triviaqa|samsum|
> |-|-|-|-|-|-|
> |0-4k|9.10|5.06|42.0|68.53|17.97|
> |4-8k|9.50|7.06|51.0|69.21|16.70|
> |8k+|8.02|6.01|38.0|65.81|20.17|
>
> # Advantage and clarification of "smaller models' performance decay faster than larger ones"
> - Firstly, this phenomenon implies FoPE is a scalable method, which should be considered as an advantage.
> - Secondly, it is not odd that larger models have better generalization.
>
> # FoPE is valuable for many reasons, although current LLMs achieve 100k+ context length
> - It is well-known that current Transformer-based LLMs achieve 100k+ context length using extrapolation as YARN [1]. **But in Fig 4, we have shown FoPE can also be used for extrapolation and has better performance than YARN.**
> - With FoPE, models achieve better length generalization trained with a much shorter context length. **Thus, FoPE can significantly improve the training efficiency of models (the longer length, the much slower training).** This is valuable for saving time and money.
>
> # Comparison with more baselines on more benchmarks
> Please check our reply to Reviewer LYp1 for detailed results.
>
> # Influence of residual connections, LayerNorm and specific activation functions
> - We have modeled the influence of activation functions in Sec 3.2.
> - Considering the residual connections and LayerNorm only deliver linear transform in frequency domain, we did not include their analysis in our paper. But for clarity, we will contain them in future revisions.
>
> # Generalization ability of FoPE on non-training data
> - **We have evaluated the OOD generalization in Sec 5.2.** (trained on Gutenberg Books and evaluated the ppl on C4)
> - Our evaluations on Sec 5.4 include datasets from SCROLLS (See Sec 5.4).
>
> # Clarification for Figure 6
> The description is clearly presented just under Fig 6 in Sec 5.5.
>
> ---
>
> [1] YaRN: Efficient Context Window Extension of Large Language Models. ICLR 2024.
>
> [2] FIRE: Functional Interpolation for Relative Positions Improves Long Context Transformers. ICLR 2024.
>
> [3] CLEX: Continuous Length Extrapolation for Large Language Models. ICLR 2024.
>
> [4] Mesa-Extrapolation: A Weave Position Encoding Method for Enhanced Extrapolation in LLMs. NeurIPS 2024.

---

### Official Review · Reviewer_LYp1 · 2025-03-17

**Overall Recommendation:** 3

**Summary:**

This paper analyses the limitations of Rotary Position Embedding (RoPE) in extending language model context length using Discrete Signal Processing theory. It identifies spectral damage from linear layers, activation functions, and insufficient frequency training as key issues affecting RoPE’s periodicity. To address this, the authors propose Fourier Position Embedding (FoPE), which constructs a Fourier Series and removes harmful frequency components to enhance length generalisation.

**Claims And Evidence:**

The claims are generally supported  and shows performance (length generalisation) gains across benchmarks. However, it does not address the computational cost of FoPE, which is important for understanding the trade-off between performance and efficiency.

**Essential References Not Discussed:**

N/A

**Experimental Designs Or Analyses:**

While the paper demonstrates performance improvements on long contexts, it doesn't provide insights into how the Fourier Position Embedding method impacts computation costs of training and inference.

**Methods And Evaluation Criteria:**

The proposed methods and evaluation criteria are appropriate. FoPE enhances length generalisation by improving position encoding, addressing spectral issues in long-context attention. The evaluation on diverse benchmarks (e.g., GovReport, MultiNews, TREC) effectively tests performance across varying context lengths. However, including more complex reasoning tasks would provide a more comprehensive assessment.

**Other Comments Or Suggestions:**

N/A

**Other Strengths And Weaknesses:**

Strengths:

1) Frequency-domain analysis of RoPE using Discrete Signal Processing (DSP), offering new insights into spectral distortions affecting long-context generalization.
2) Practical impact with Fourier Position Embedding (FoPE), demonstrating strong improvements in context extension across model scales and tasks.

Weaknesses:

1) Computational cost of FoPE is unclear, especially for large-scale models.

**Questions For Authors:**

1) Could the authors include benchmarks on reasoning tasks like MMLU and GSM8K to demonstrate greater task diversity?

2) The authors are asked to provide a comparison of computational analysis.

**Relation To Broader Scientific Literature:**

This paper extends Rotary Position Embedding (RoPE) by analyzing its frequency-domain properties using Discrete Signal Processing (DSP) and identifying spectral distortions that hinder long-context generalization. Building on prior work in positional embeddings (Su et al., 2021; Press et al., 2022) it introduces Fourier Position Embedding (FoPE), which filters harmful frequency components to improve attention’s periodicity.

**Theoretical Claims:**

No.

---

> ### Author Rebuttal · Authors · 2025-03-29
>
> Thanks for the detailed comments from the reviewer, we would like to make several clarification below.
>
> # Computation cost of FoPE is similar to RoPE
> We agree with the reviewer that the efficiency and computation cost are essential for Position Embedding, thus:
> - FoPE keeps the rotary matrix having the similar shape as RoPE.
> - FoPE keeps the similar pipeline as RoPE: ① pre-compute the rotary matrix and save the matrix in cache before training; ② take out the matrix for rotation during training.
>
> Therefore, FoPE is as efficient as RoPE, independent of the model scale. (Reviewer FqQv also acknowledges that FoPE is lightweight and effective in "Other Strengths And Weaknesses")
>
> # Evaluation on more tasks and baselines
> It is a nice suggestion to evaluate FoPE on more diverse tasks, thus we supplement the following experiments:
> - We validate the effectiveness of FoPE by evaluating on 10+ more benchmarks and comparing with more 3 more baselines. These supplementary experiments are conducted on OLMo-1.2B training on C4 datasets, keeping all of the settings similar to the experiments in Sec 5.2 (line 293-329). We do not show the results on GSM8K as all methods achieve nearly 0 acc, which may be caused by the lack of math data in C4.
>
> | methods | avg acc | basic_arithmetic | social_iqa | winogrande | openbook_qa | sciq  | hellaswag | piqa | commonsense_qa | arc_easy |
> |-|-|-|-|-|-|-|-|-|-|-|
> | NoPE | 42.14 | 25.67 | 43.71 | 51.86 | 29.80 | 76.70 | 41.83 | 68.83 | 31.61 | 51.40 |
> | ALiBi | 42.93 | 24.97 | 42.53 | 53.12 | 31.40 | 77.70 | 43.06 | 69.42 | **33.42** | **53.68** |
> | KERPLE | 43.22 | 25.03 | 43.81 | **54.07** | 32.40 | **78.20** | 43.65 | 69.64 | 32.92 | 52.46 |
> | FIRE | 42.38 | 25.60 | 42.63 | 49.88 | 33.40 | 77.00 | 42.75 | 69.31 | 32.92 | 50.35|
> | RoPE | 42.98 | 24.60 | 43.45 | 51.54 | **33.60** | 77.10 | 43.36 | **70.13** | 33.01 | 52.98 |
> | FoPE | **43.37** | **26.17** | **44.12** | 53.20 | 32.20 | 77.80 | **43.83** | 70.08 | 32.92 | 53.33 |
>
> | methods | avg acc | mmlu_stem | mmlu_social_sciences | mmlu_humanities | mmlu_other |
> |-|-|-|-|-|-|
> | NoPE | 25.81 | 25.47 | 25.38 | 28.01 | 24.39 |
> | ALiBi | 25.99 | 25.93 | 27.20 | 26.44 | 24.39 |
> | KERPLE | 26.16 | 26.02 | 25.97 | 27.82 | 24.82 |
> | FIRE | 26.04 | 26.34 | **27.41** | 26.25 | 24.16 |
> | RoPE | 26.68 | 27.01 | 27.26 | 27.87 | 24.53 |
> | FoPE | **27.57** | **27.30** | 27.31 | **29.89** | **25.79** |
>
> | methods | avg ce loss | natural_qs_open | trivia_qa_wiki | arc_easy |
> |-|-|-|-|-|
> | NoPE | 1.4513 |1.4334 | 1.6129 | 1.3077 |
> | ALiBi | 1.4242 |1.3879 | 1.6057 | 1.2789 |
> | KERPLE | 1.4284 | 1.4149 | 1.5878 | 1.2825 |
> | FIRE | 1.4580 | 1.4365 | 1.6258 | 1.3118 |
> | RoPE | 1.4225 | 1.4114 | 1.5973 | 1.2588 |
> | FoPE | **1.3941** | **1.3818** | **1.5736** | **1.2272** |
>
> - **Although FoPE is primarily designed for length generalization and long-context tasks, it also performs comparably or even surpasses all baseline methods**. But on MMLU, all methods have weak performance, perhaps caused by the training data. In the future, we may conduct more experiments on datasets containing reasoning and math data.
> - We also conduct evaluation similar to those in Sec 5.2 (See https://anonymous.4open.science/r/FoPE-Supplementry-Material/Supplementary_Main_Experiments.pdf). **FoPE still demonstrates the best accuracy on passkey retrieval and the second-best ppl on C4 (only slightly behind ALiBi).**

---

### Official Review · Reviewer_FqQv · 2025-03-23

**Overall Recommendation:** 3

**Summary:**

## update after rebuttal

I read the latest clarification by the authors, and understand q and k in Re[qk*e^{i{m-n}\theta}] are not exactly the 2-dim vector [q_x, q_y]^T but an implicit complex number. I raised my score back.
******

This paper points out that in RoPE, different dimensions correspond to different frequencies,
and suggests viewing the interaction between queries and keys from the perspective of non-Uniform DFT (Eq.2 - Eq.4).
(However, there is an unclear point here: while RoPE is motivated by complex number rotation, its implementation is actually based on vector rotation--—they are not equivalent. The authors' analysis is based on the former, but the actual analysis object is the latter, and the authors' implementation is also the latter. Therefore, at minimum, they should explain this gap.)

Based on this **deconstructive analysis**,
the authors identify two issues:
1. If we consistently view RoPE from a signal perspective, we cannot ignore the spectral leakage caused by activation functions and linear layers;
2. Considering that low frequencies correspond to stable components in long contexts, while the text used in training is not very long, these low-freq components are insufficiently trained.

The authors propose two improvements to RoPE:
1. Allow feature dimensions that previously corresponded to a *single* frequency to now correspond to *multiple* frequencies, while still maintaining a *primary* frequency, enabling more flexible adjustment of the disturbed spectral information.
2. Set the insufficiently trained dimensions directly to 1.

In experiments, the authors primarily validate the benefits of FoPE for length extrapolation capabilities. Experimental scenarios include pre-training, continual pre-training, and fine-tuning.

**Claims And Evidence:**

I have concerns about the following claim:

The authors view the interaction between queries and keys from the perspective of non-Uniform DFT (Eq.2 - Eq.4),
and they hope this perspective can be maintained throughout the model, thus proposing two improvements.
The starting point of these approaches is understanding RoPE's formula as Eq.2.
However, **it should be noted that RoPE's actual implementation is not Eq.2.**
Eq.2 can be considered as the heuristic starting point of RoPE,
while the actual implementation rotates each pair of dimensions by a certain angle, which is not equivalent to Eq.2,
therefore, in fact **we cannot write a strict Fourier form equation**.

It can be said that:
**RoPE has both a "heuristic approach" and a "practical approach",
and the authors' analysis is based on the former "heuristic approach" but improves upon the "practical approach".**

I would like to hear other reviewers' and AC's opinions on this point.

**Essential References Not Discussed:**

Not sure.

**Experimental Designs Or Analyses:**

Yes.

PS: I have not personally conducted similar experiments and have insufficient understanding of the experimental details.

**Methods And Evaluation Criteria:**

Yes.

**Other Comments Or Suggestions:**

What is the typical range of values for $\omega_m = 1/{\theta}^{(2m/M)}$ in Eq.2? This should be explicitly stated.

**Other Strengths And Weaknesses:**

Other Strengths:
1. Provides a deconstructive analysis of existing models；
2. The proposed method is lightweight and effective；
3. Although I am not very familiar with the experimental approaches in this direction, I find the authors' writing of the experimental section to be very fluent.

Other Weaknesses:
- I still have concerns about the signal-based explanation. First, there are the concerns raised in (Claims And Evidence). Additionally, I am concerned about: in the authors' analysis, attention with RoPE becomes an inverse Fourier transform form, which essentially treats the input as a frequency domain response. However, if this is the case, how can we analyze multiple stacked blocks from a signal perspective?

**Questions For Authors:**

Please refer mainly to the (Claims And Evidence) section,
and also see the (Other Weaknesses) and (Other Comments Or Suggestions) sections

**Relation To Broader Scientific Literature:**

This paper provides insights into understanding Transformer architecture and solving long-range dependency problems from a signal analysis perspective.

**Theoretical Claims:**

There are no specific theoretical claims, but the paper's analysis has certain theoretical aspects. Please see my concerns in (Claims And Evidence).

---

> ### Author Rebuttal · Authors · 2025-03-29
>
> Thanks for the insightful comments from the reviewer, we are going to make clarifications for the concerns above.
>
> # Clarification for the "heuristic approach" and "practical approach" of RoPE/FoPE
> It seems this is the major concern of the reviewer, we acknowledge the importance of clarifying this point:
>
> 1. **The Rotary Matrix implies the real part of the complex number in Matrix Space, which is also the main difference between the "heuristic approach" and "practical approach" of RoPE.** The derivation is in Sec 3.4 of Roformer's original paper [1], we also provide a derivation under our understanding:
> In the 2D case of RoPE, the vectors $q$ and $k$ can be viewed either as 2D plane vectors ($\mathbf{q}=(q_x,q_y)^T$) or as vectors in the complex plane ($\mathbf{q}=\|\mathbf{q}\|e^{i\theta_q}$, $\|q\|=\sqrt{q_x^2+q_y^2}$, $\theta_q = \arctan{\frac{q_x}{q_y}}+k\pi$ ). The inner product of the two vectors is equal to $\langle\mathbf{q},\mathbf{k}\rangle=Re[\mathbf{qk}^*]$ (the proof is at the end of this reply).
>
> 2. **As taking the real part does not change the frequency-domain property of a vector, thus: ① RoPE's actual implementation still achieves Eq.2; ② we can still write a strict Fourier form equation.**
>
> 3. **As a result, our theoretical analysis and implementation are well-aligned.** To provide better clarity, we will explicitly include a derivation in the future revisions.
>
> 4. **Our further empirical results also demonstrate that the actual implementation of RoPE properly achieves its theoretical motivation, also well-aligning with our frequency-domain analysis (See https://anonymous.4open.science/r/FoPE-Supplementry-Material/Empirical_Analysis_on_Spectrum_Damage.pdf).** In this experiment, we force every token only has one frequency component in the first layer. Then, we compare the spectrum of the first and second layer with DFT on attention map (aligned with the derivation in Sec 2.2).
>
> # Relationship between fourier/signal modeling and multiple stacked blocks
>
> - Firstly, we want to clarify that, **as the Fourier Transform is a linear transform [2], it does not change the main properties of models.** Thus, the multiple stacked blocks are still in charge of modeling more diverse/complex and higher-order functions/signals, which improves the expression ability of models.
> - Secondly, based on our modeling from a signal/frequency-domain perspective, **different layers actually process signals of different frequency distribution.** This is useful for improving the expression ability of models. However, **RoPE wrongly regards the frequency distribution as same in different layers, leading to drawbacks in length generalization**.
> - Lastly, our modeling in frequency-domain can analyze the periodicity of attention mechanism, providing insights into the drawback of RoPE in length generalization.
>
> # Typical range of values for $\omega_m$
> We have shown these statistics in Appendix D.3 (line 727-728). The frequencies of RoPE are between (0, 1] and samples more densely near 0.
>
> # Comparison with more baselines on more benchmarks
> We also conduct more supplementary experiments, please check our reply to Reviewer LYp1 for detailed results.
>
> ---
> ## Reference
> [1] Jianlin Su, et al. RoFormer: Enhanced Transformer with Rotary Position Embedding.
>
> [2] Oppenheim, et al. Signals and Systems.
>
> ---
> ## Additional Proof
> $$\langle f_q(\mathbf{x_m},m),f_k(\mathbf{x_n},n)\rangle$$
> $$=\langle \mathbf{q}e^{im\theta},\mathbf{k}e^{in\theta}\rangle$$
> $$=\text{Re}[\mathbf{qk}^*e^{i(m-n)\theta}]$$
> $$=\text{Re}[\|q\|\|k\|e^{i[\theta_q-\theta_k+(m-n)\theta]}]$$
> $$=\|q\|\|k\|\cos(\theta_q-\theta_k+(m-n)\theta)$$
> $$=\|q\|\|k\|\left[ \cos(\theta_q-\theta_k)\cos(m-n)\theta - \sin(\theta_q-\theta_k)\sin(m-n)\theta\right]$$
> $$=\|q\|\|k\|\left[(\cos\theta_q\cos\theta_k+\sin\theta_q\sin\theta_k)\cos(m-n)\theta-(\sin\theta_q\cos\theta_k-\cos\theta_q\sin\theta_k)\sin(m-n)\theta \right]$$
> $$=(q_xk_x+q_yk_y)\cos (m-n)\theta - (q_yk_x-q_xk_y)\sin (m-n)\theta$$
> $$=[q_x, q_y][\cos (m-n)\theta, -\sin (m-n)\theta; \sin (m-n)\theta, \cos (m-n)\theta][k_x, k_y]^T$$
> $$=[q_x, q_y][\cos m\theta, \sin m\theta; -\sin m\theta, \cos m\theta][\cos n\theta, -\sin n\theta; \sin n\theta, \cos n\theta][k_x, k_y]^T$$
> $$=(\mathbf{R_{\theta,m}q})^T(\mathbf{R_{\theta,n}k})$$
> $$=\mathbf{q^T R_{\theta,m-n}k}$$

---

> > ### Comment · Reviewer_FqQv · 2025-04-06
> >
> > > Re: Clarification for the "heuristic approach" and "practical approach" of RoPE/FoPE
> >
> > **The "heuristic approach" and "practical approach" of RoPE are indeed not equivalent, as we can clearly see from your Additional Proof derivation.**
> >
> > In this derivation, you attempted to derive the "practical approach" formula from the "theoretical heuristic approach".
> >
> > The third line
> > $$\textrm{Re}[{\mathbf{q}\mathbf{k}^{*}}e^{\mathrm{i}(m-n)\theta}]$$
> >
> > is correct, but the next line is incorrect. Here, $\mathbf{q}\mathbf{k}^{*}$ is a real number, equal to
> > $\|\mathbf{q}\| \|\mathbf{k}\| \cos(\theta_q - \theta_k)$,
> > or it can be written as
> > $\mathrm{Re}[\|\mathbf{q}\| \|\mathbf{k}\| \mathrm{e}^{i(\theta_q - \theta_k)}]$. Therefore, after the third line, you can only write:
> >
> > $$
> > \mathrm{Re}[ \underbrace{\mathrm{Re}[\|\mathbf{\theta_q}\| \|\mathbf{\theta_k}\| \mathrm{e}^{i(\theta_q - \theta_k)}]}_{\textrm{a real number}} \mathrm{e}^{\mathrm{i}(m-n)\theta}],
> > $$
> >
> > And this is not equal to the fourth line's formula. The difference lies in whether you accidentally let $\theta_q$ and $\theta_k$ affect the angle in the complex plane.
> >
> > Therefore, your "practical approach" and "theoretical heuristic approach" are not equivalent.
> >
> > I'm not saying this to negate your contribution - even at the heuristic level, your analysis and attempts are meaningful.
> >
> > **However, there is indeed an overclaim here, and it's quite crucial, so I will discuss this carefully with the AC and other reviewers.**
> >
> > Additionally, **there seem to be another overclaim** in the paper.
> > The authors analyze RoPE from the perspective of NUDFT, but according to Wikipedia [https://en.wikipedia.org/wiki/Non-uniform_discrete_Fourier_transform], the number of frequencies in NUDFT needs to reach N.
> > This point should also be clarified.
> >
> > > Re: Relationship between fourier/signal modeling and multiple stacked blocks
> >
> > I didn't quite understand this part.
> >
> > What I mean is, since the authors want to view the Transformer from a signal perspective throughout,
> > can the output of an attention layer with RoPE be considered as the result of inverse NUDFT? If so, how should this be connected with the subsequent V and more self-attention layers?

---

> > > ### Author Response · Authors · 2025-04-06
> > >
> > > We appreciate the reviewer's elaborate comments on our derivation. But there are several essential clarification we want to present:
> > >
> > > > Re: $\mathbf{qk^*}$ is real number, equal to $|q||k|\cos(\theta_q-\theta_k)$ (or $Re[|q||k|e^{i(\theta_q-\theta_k)}]$
> > >
> > > We would clarify two crucial misunderstandings:
> > > - $\mathbf{q}$, $\mathbf{k^*}$ and $\mathbf{qk^*}$ are all complex numbers, but not real numbers.
> > > - In the third line of our formula, it is $Re[\mathbf{qk^*}e^{i(m-n)\theta]}$, but not $Re[<\mathbf{q, k^*}>e^{i(m-n)\theta}]$.
> > >
> > > Thus, **the third line of our formula calculates the "multiplication" but not the "inner product" between $\mathbf{q}$ and $\mathbf{k}$.** This would lead to a totally different results, as $\mathbf{qk^*} = |q||k|e^{i[\theta_q-\theta_k]}$, but $<\mathbf{q, k^*}> = Re[\mathbf{q(k^*)^*}]=Re[\mathbf{qk}]=|q||k|\cos(\theta_q+\theta_k)$.
> > >
> > > We understand that the reviewer want to present the "inner product" of $q$ and $k$ in attention mechanism, but the "inner product" have been presented in the second line of our derivation $<\mathbf{q}e^{im\theta}, \mathbf{k}e^{in\theta}>$.
> > >
> > > **Overall, we are still confident that our derivation is correct. We welcome further discussion with the Reviewer FqQv, as well as the other reviewers and ACs.**
> > >
> > > Additionally, we would clarify some ambiguous definition in our first version rebuttal. That is, the original format of a complex number should be $\mathbf{q}=|q|e^{i\theta_q}$ ($|q|=\sqrt{q_x^2+q_y^2}, \theta_q=\arctan{\frac{q_x}{q_y}}+k\pi$), while the vector format of a complex number should be $\mathbf{\vec{q}}=[q_x, q_y]^T$. Accordingly, we rewrite the derivation at the end of this reply (without changing any of its original meaning).
> > >
> > > > Re: Your "practical approach" and "theoretical heuristic approach" are not equivalent.
> > >
> > > We would like to respectfully clarify that, **we did not claim the equivalence between the "practical approach" and "theoretical approach" of RoPE/FoPE**. The expressions we used are "RoPE **implicitly achieves** Periodic Attention based on NUDFT" and "the practical approach and theoretical approach of RoPE/FoPE are **well-aligned**". **These expressions are deliberately chosen and clearly distinct from claiming strict equivalence.**
> > >
> > > **Also, our derivation explains the actual relationship between the "practical approach" and "theoretical approach" of RoPE/FoPE.** That is, these two approaches are geometrically aligned. In implementation, RoPE transforms complex-number operations in the complex plane into vector operations in the 2D real plane.
> > >
> > > > Re: The number of frequencies in NUDFT needs to reach N, this point should also be clarified.
> > >
> > > The number N is not essential for NUDFT, and can be set to any reasonable number. This can be supported by the official implementation of NUDFT from MATLAB (https://www.mathworks.com/help/matlab/ref/double.nufft.html) and other repo (https://mind-inria.github.io/mri-nufft/nufft.html).
> > >
> > > > Re: Can the output of an attention layer with RoPE be considered as the result of inverse NUDFT? If so, how should this be connected with the subsequent V and more self-attention layers?
> > >
> > > We did not claim that "**the output of an attention layer with RoPE** could be considered as the result of inverse NUDFT", but we claim that "**the attention score/weight of a self-attention with RoPE (without softmax)** could be considered as the result of inverse NUDFT". **This is also be claimed by the title in our paper, this transform implicitly achieves a Periodic Attention, which have the potential to be extended periodically for better length generalization.**
> > >
> > > Thus, the subsequent V is a weighted average of the former V, where the weight is the **Periodic Attention Score**. Still, the V vector implies the coefficients of frequency components.
> > >
> > > As for the influence of passing coefficients across layers, we have answered in our former rebuttal. That is, "the multiple stacked blocks are still in charge of modeling more diverse/complex and higher-order functions/signals, which improves the expression ability of models."
> > >
> > > ---
> > >
> > > **Additional Derivation**
> > >
> > > $< f_q(\mathbf{x_m},m),f_k(\mathbf{x_n},n)>$
> > >
> > > $=\langle \mathbf{q}e^{im\theta},\mathbf{k}e^{in\theta}\rangle$
> > >
> > > $=Re[\mathbf{qk}^*e^{i(m-n)\theta}]$
> > >
> > > $=Re[\|q\|\|k\|e^{i[\theta_q-\theta_k+(m-n)\theta)]}]$
> > >
> > > $=\|q\|\|k\|\cos(\theta_q-\theta_k+(m-n)\theta)$
> > >
> > > $=\|q\|\|k\|\left[ \cos(\theta_q-\theta_k)\cos(m-n)\theta-\sin(\theta_q-\theta_k)\sin(m-n)\theta\right]$
> > >
> > > $=\|q\|\|k\|\left[(\cos\theta_q\cos\theta_k+\sin\theta_q\sin\theta_k)\cos(m-n)\theta
> > > -(\sin\theta_q\cos\theta_k-\cos\theta_q\sin\theta_k)\sin(m-n)\theta \right]$
> > >
> > > $=(q_xk_x+q_yk_y)\cos(m-n)\theta - (q_yk_x-q_xk_y)\sin(m-n)\theta$
> > >
> > > $=[q_x, q_y] [\cos(m-n)\theta, -\sin(m-n)\theta; \sin(m-n)\theta, \cos(m-n)\theta] [k_x, k_y]^T$
> > >
> > > $=[q_x, q_y] [\cos m\theta, \sin m\theta; -\sin m\theta, \cos m\theta][\cos n\theta, -\sin n\theta;\sin n\theta, \cos n\theta][k_x, k_y]$
> > >
> > > $=\mathbf{\vec{q}}^T\mathbf{R}_{\theta,m-n}\mathbf{\vec{k}}$

---

### Decision · Program_Chairs · 2025-05-01

**Decision:**

Accept (poster)

**Comment:**

This paper uses insights from discrete signal processing to analyze how RoPE enables periodic attention patterns and its limitations in that regard, identified as a form of "spectral damage". The paper proposes FoPE as a solution, which is differs from RoPE in which it models each dimension not as a single frequency but as a Fourier Series, consisting of a dominant frequency component and several harmonic components. Experiments show that FoPE maintains a better performance for increased context lengths.

This paper provides a valuable contribution which analyzes weaknesses of current models and uses insights from discrete signal processing to propose a solution. Reviewers point out as weaknesses concerns about the signal-based explanation and the computational cost (both of which were clarified in the rebuttal), as well as somewhat inconclusive experiments and results, which does not completely demonstrate the value of FoPE versus RoPE or other embedding methods. The authors addresses most of the concerns. I am leaning towards acceptance.